# Do Bias Mitigation Methods Generalize? A Cross-Modality Study

## Abstract

Spurious correlations, defined as predictive but non-causal relationships within training data, constitute a significant challenge for deep learning. When such shortcuts exist in a dataset, models tend to exploit them instead of learning the intended, task-relevant features, resulting in biased predictions and poor generalization. Although numerous bias mitigation methods have been developed, they are primarily evaluated on natural images, their ability to generalize to other modalities and domains, namely text (e.g., occupational gender imbalance and lexical bias), audio (e.g., demographic disparities and device signatures), medical imaging (e.g., hospital-level biases such as scanner or protocol differences), and video (e.g., scene background bias), remains largely unexplored. In this work, we conduct the first comprehensive cross-modality benchmark study, evaluating 14 bias mitigation methods across 6 datasets spanning text, audio, medical imaging, and video. For each dataset, we introduce tailored configurations designed to assess bias mitigation performance. Our findings show that several methods provide consistent improvements across modalities, with a subset exhibiting statistically significant bias mitigation in all domains. This study offers the first systematic evidence of cross-modal generalization for bias mitigation approaches and establishes a benchmark resource aimed at encouraging the development of bias mitigation methods that extend beyond the natural images domain. Code and data will be released publicly upon acceptance.

## 1 Introduction

Deep learning systems have become ubiquitous across society, driving state-of-the-art performance in countless applications. As these models are increasingly deployed to make high-stakes decisions affecting individuals and populations (Melzi et al., 2024; Sarridis et al., 2023; Moussa et al., 2025), understanding and mitigating their inherent limitations is paramount for developing trustworthy Artificial Intelligence (AI) systems.

As such systems increasingly rely on large and heterogeneous datasets, they inevitably reflect the statistical characteristics and imperfections present in the data. This can introduce various forms of bias into the learned representations and decision-making processes, influencing model behavior in ways that are not always aligned with the intended task. In particular, high co-occurrence between certain data attributes and target labels that does not reflect true causal relationships, namely spurious correlations, represents one of the most significant challenges to robust generalization and bias mitigation in deep learning systems (Geirhos et al., 2020; Shah et al., 2020). When neural networks encounter training data containing such non-causal associations, they tend to learn these superficial patterns due to their easier-to-learn nature (Shah et al., 2020; Yang et al., 2024a; Sarridis et al., 2025b). In particular, rather than discovering the genuine causal mechanisms underlying a task, models exploit shortcuts and contextual cues that happen to correlate with labels in training data (Geirhos et al., 2020; Saranrittichai et al., 2022). This fundamental failure mode creates profound practical implications, i.e., models that perform accurately within training distributions often fail catastrophically when deployed in new environments where these spurious correlations cease to hold (Mulchandani & Kim, 2025; Bassi et al., 2024).

Over the past several years, the research community has developed numerous bias mitigation methods and evaluated them extensively on vision benchmarks (Erobic et al., 2022; Sarridis et al., 2025b; Wu et al., 2023; Sarridis et al., 2025a). This body of work has documented diverse manifestations of spurious correlations in vision datasets such as ImageNet (Geirhos et al., 2018), CelebA (Seo et al., 2022a), and Waterbirds (Sagawa et al., 2019), where models tend to learn texture rather than shape (Geirhos et al., 2018), exploit background contextual cues instead of foreground objects (Sagawa et al., 2019; Sadiku, 2025), or rely on demographic correlations to achieve task performance (Seo et al., 2022a; Serianni et al., 2025). These vision-centric benchmarks have substantially advanced our understanding of bias in deep learning.

Beyond vision, deep learning systems operate across multiple modalities and domains – Natural Language Processing (NLP) (Liusie et al., 2022; Zhou et al., 2024), audio (Oglic et al., 2022) and video (Zang et al., 2023) systems are prevalent in several AI applications, while systems processing medical images (Saab et al., 2022) play a central role in many healthcare applications. Biases manifest distinctly across modalities, arising from different origins. For instance, in text modalities, spurious correlations can emerge due to lexical patterns, i.e., associating identity terms (such as "muslim", "black", or "gay") with toxicity labels (Garg et al., 2019; Yadav et al., 2022). In audio, especially speech recognition systems, severe demographic biases arise from training data imbalance that systematically under-represent marginalized dialects and speech characteristics (Koenecke et al., 2020). Beyond demographic biases, audio datasets exhibit spurious correlations with task-irrelevant cues such as recording device signatures, microphone characteristics, and background noise patterns that happen to correlate with labels (Solewicz et al., 2016). In medical imaging, dataset biases can arise from institutional and technical variations, i.e., medical data comes from diverse sources (different hospitals, scanners, manufacturers, acquisition protocols), introducing systematic differences in image appearance that neural networks exploit rather than learning genuine diagnostic markers (Fortin et al., 2018; Gadewar et al., 2024). Finally, in video understanding, models tend to rely on scene context and background cues rather than genuine target signals, i.e., action recognition models in video datasets exploit the spatial clustering of actions (basketball on courts, cooking in kitchens) as shortcuts (Chung et al., 2022; Li et al., 2023). In several of these alternative domains, work on addressing spurious correlations remains sparse, fragmented, and often highly task-specific (Qin et al., 2024; Stanley et al., 2025).

Despite the modality-agnostic nature of spurious correlations, the generalization capabilities of bias mitigation approaches beyond natural images remain largely unexplored. To address this gap, this work introduces the first comprehensive cross-modality empirical study that systematically evaluates whether bias mitigation methods originally designed for natural images generalize to other modalities and domains. We benchmark 14 established and recent bias mitigation methods across 6 datasets, including text, audio, medical imaging, and video. Our study is framed as a benchmark effort, explicitly designed to provide researchers with systematic evidence on method generalization capabilities and to encourage the community to develop bias mitigation approaches that perform robustly across diverse modalities.

Specifically, the three central research questions guiding this work are:
**RQ1:** Do bias mitigation methods originally tested for natural image classification reduce bias in other domains/modalities (text, audio, medical imaging, video)?
**RQ2:** How do different methods rank, and are method rankings maintained across different domains/modalities compared to natural image benchmarks?

By systematically answering these questions, we provide the first empirical evidence on the generalization and limitations of bias mitigation approaches in highly diverse scenarios.

## 2 Background

Spurious correlations in machine learning occur when training datasets contain high statistical association between task-irrelevant attributes and target labels, enabling models to achieve acceptable training performance by exploiting these correlations rather than learning genuine causal relationships (Geirhos et al., 2020; Ye et al., 2024). Understanding the origins of such correlations requires examining both the statistical properties of datasets and the optimization dynamics of neural network training (Fabbrizzi et al., 2022; Sarridis et al., 2025a). The phenomenon of prioritizing easy-to-learn features in neural network optimization provides a

theoretical foundation for understanding spurious correlations (Shah et al., 2020; Yang et al., 2024a; Sarridis et al., 2025b). When training data contains both causal and spurious features that enable task performance, networks exploit whichever feature (causal or spurious) has less complexity (Sarridis et al., 2025b;a; Shah et al., 2020; Vasudeva et al., 2023). Crucially, spurious features are often substantially simpler than genuine causal features—for example, background texture or color statistics versus object shape—making models default to these shortcuts when both predict the target label equally well in training data (Geirhos et al., 2018; Hermann et al., 2020). Sampling, measurement, and historical biases during data collection are the main mechanisms through which spurious correlations emerge (Fabbrizzi et al., 2022; Suresh & Guttag, 2021). Selection bias occurs when data collection fails to randomly sample from the target population, instead systematically excluding or under-representing particular demographic groups or task-relevant conditions (Fabbrizzi et al., 2022; Chauhan et al., 2025). Historical bias reflects societal biases embedded in data collected from past practices, where discriminatory historical patterns become encoded in datasets (Suresh & Guttag, 2021; Australian Human Rights Commission, 2020). Measurement bias arises when data collection methods or annotation procedures introduce systematic distortions, such as when human annotators introduce subjective biases during labeling (Haliburton et al., 2023; Suresh & Guttag, 2021).

## 2.1 Bias in Images

### 2.1.1 Natural Images

Natural image datasets, such as ImageNet (Deng et al., 2009), CelebA (Liu et al., 2015), and Waterbirds (Sagawa et al., 2019), have become foundational benchmarks for spurious correlation research and have enabled substantial progress in understanding and mitigating dataset biases in computer vision.

ImageNet remains the most widely used large-scale benchmark in computer vision and has significantly shaped the study of spurious correlations and debiasing methods. Systematic work has revealed that models trained on ImageNet exhibit a marked texture bias. Specifically, in controlled settings where the shape and texture cues of objects are artificially separated, standard convolutional neural networks (CNNs) like ResNet-50 overwhelmingly categorize ambiguous objects by texture rather than shape—contrary to the human vision system, which primarily relies on shape information (Geirhos et al., 2018). This demonstrates a fundamental misalignment between deep neural representations and human perception, as networks exploit textural patterns and local statistics that correlate with labels in the training data, rather than learning truly semantic object identities (Hermann et al., 2020). Other research has also highlighted that, beyond texture, models trained on ImageNet and similar datasets frequently depend on task-irrelevant background cues for prediction. For instance, models retain surprisingly high recognition accuracy even when provided with background regions alone, and can be misled by background-foreground mismatches (Xiao et al., 2021). To facilitate the study of robustness against real-world spurious correlations and corruptions, datasets like ImageNet-C (Hendrycks & Dietterich, 2019) have been designed to systematically evaluate model vulnerability to common corruptions (e.g., noise, blur, weather) and perturbations. Results indicate that, despite progress in model architecture, classifier robustness to such natural distribution shifts has improved only marginally, with simpler models and more complex models alike struggling with spurious signals introduced by corruptions (Hendrycks & Dietterich, 2019).

When such spurious correlations involve protected attributes—such as gender, race, or age—they create fairness issues that can perpetuate discrimination and inequitable outcomes. The CelebA dataset, which contains celebrity face images annotated with 40 demographic and appearance attributes (Liu et al., 2015), exemplifies how real-world datasets encode spurious correlations between certain attributes and demographic groups. For instance, substantial correlations exist between female gender and appearance attributes such as hair color, earrings, lipstick, and heavy makeup (Sarridis et al., 2025b), patterns that emerged naturally from the dataset's composition process rather than reflecting any causal relationship. Models trained to perform seemingly neutral tasks, such as predicting hair color on CelebA, implicitly learn to exploit gender as a proxy classifier, achieving high overall accuracy while demonstrating severe performance disparities across demographic groups (Hong & Yang, 2021).

### 2.1.2 Medical Images

Confounding biases from systematic institutional and technical variations represent a major source of dataset bias in medical imaging. Medical imaging data comes from diverse sources—different hospitals, clinics, and imaging centers—each using potentially different scanner hardware (varying manufacturers, field strengths, and hardware versions), acquisition protocols, and image reconstruction algorithms (Galanty et al., 2024; Koçak et al., 2025). These technical variations introduce systemic differences in image appearance that can be reliably detected by trained neural networks. Scanner-specific confounds have been documented across multiple medical image types, including MRI, where biases arise from scanner manufacturer, scanner upgrade history, scanner field strength variations, scanner drift over time, gradient nonlinearity hardware variations, and subject positioning (Fortin et al., 2018; Gadewar et al., 2024). These biases can influence quantitative imaging metrics substantially, affecting the reliability of models trained on multi-site, multi-scanner datasets.

A second large body of work has examined whether deep models for medical imaging exploit demographic attributes (sex, age, race) as shortcuts. Models trained on chest X-rays have been shown to predict patient race from images alone with accuracy far exceeding human radiologists (Gichoya et al., 2022), and disease classifiers have been observed to systematically underdiagnose specific subpopulations (Seyyed-Kalantari et al., 2021). This has motivated a line of work on demographic-fairness-aware training for medical imaging (Zong et al., 2023; Lin et al., 2023). However, such biases differ fundamentally from the types of biases considered in other settings. In medical contexts, demographic attributes can be clinically relevant and may legitimately contribute to accurate diagnosis (Yang et al., 2024b). As a result, removing their influence entirely, unlike in cases where the bias source is clearly irrelevant (e.g., dataset origin), may be neither appropriate nor desirable. Moreover, demographic annotations are often noisy, incomplete, or inconsistently defined, further complicating their use (Ricci Lara et al., 2022). Thus, in this study, we focus on acquisition/data-source bias in medical images, which is clearly defined and allows for reliable conclusions.

## 2.2 Bias in Text

NLP models exhibit spurious correlations at multiple levels of linguistic organization, from token-level associations to concept-level patterns that cause models to exploit misleading correlations between superficial linguistic features and task labels (Wu et al., 2022; Yang et al., 2023).

Natural Language Inference datasets such as Stanford Natural Language Inference (SNLI) (Bowman et al., 2015) and Multi-Genre Natural Language Inference (MultiNLI) (Williams et al., 2018) have become prominent examples of spurious correlation problems in NLP. Models trained on these datasets exploit spurious correlations between surface-level lexical patterns and labels rather than learning genuine semantic relationships. For instance, models learn that the presence of certain keywords (such as "not", "people", or high word overlap) correlates with specific entailment decisions, enabling acceptable performance on training data while failing when these surface patterns are removed (Wu et al., 2022; Rajaee et al., 2022). Specifically, Wu et al. (2022) (Wu et al., 2022) demonstrated that models trained on standard SNLI and MultiNLI fail dramatically on out-of-distribution test sets where spurious correlations are not present.

Concept-level spurious correlations manifest as inappropriate associations between semantic concepts and labels. Language models trained on text corpora reflecting historical occupational gender imbalance learn to associate particular professions with specific genders (Kotek et al., 2023; Jiang et al., 2025). When prompted to generate text about a nurse, models overgenerate female pronouns; when generating about an engineer, models overgenerate male pronouns (Kotek et al., 2023).

Text classification tasks exhibit token-level spurious correlations where models learn unintended biases from data imbalance. Toxic comment classification systems trained on imbalanced corpora learn to associate identity terms (such as "gay", "muslim", or "black") with toxicity labels due to their overrepresentation in toxic examples (Garg et al., 2019; Yadav et al., 2022). This causes models to mislabel non-toxic comments containing identity terms as toxic. Counterfactual data augmentation approaches that generate examples with replaced identity terms while preserving syntax have been proposed to reduce these spurious correlations, though their effectiveness varies across datasets and domains (Yadav et al., 2022).

### 2.3 Bias in Audio

Audio datasets, particularly those used for Automatic Speech Recognition (ASR), exhibit severe demographic biases and demonstrate clear instances of spurious correlations in the speech domain. Systematic evaluations of major commercial ASR systems (Amazon, Apple, Google, IBM, Microsoft) have revealed dramatic performance disparities across demographic groups and dialects (Koenecke et al., 2020).

Acoustic modeling bias and dialectal variation constitute primary mechanisms by which ASR systems fail for speakers of marginalized dialects (Koenecke et al., 2020). Training data imbalance in speech corpora constitutes the root cause of ASR spurious correlations. The majority of training examples represent majority-dialect, controlled, formally-spoken utterances, lacking representation of minority dialect acoustic patterns and spontaneous speech characteristics. This imbalance is compounded by the historical underrepresentation of African Americans and other marginalized groups in linguistic datasets, which systematically exclude or underrepresent non-mainstream dialects and conversational speech styles, perpetuating spurious correlations between demographic attributes and speaking style (Koenecke et al., 2020).

Beyond demographic biases, audio datasets exhibit spurious correlations with task-irrelevant acoustic and environmental cues. Models may rely on recording device signatures, background noise patterns, microphone characteristics, or channel artifacts that happen to correlate with labels in training data but do not generalize. For example, a speaker recognition system trained on recordings from a particular device may exploit device-specific acoustic signatures rather than learning speaker identity, leading to failure when tested on audio from different microphones or recording environments (Solewicz et al., 2016).

As in other modalities, the fundamental problem underlying these spurious correlations is that neural networks exploit whichever features are simplest and most readily available in the training data—often task-irrelevant acoustic or demographic cues—when they correlate with labels equally well as genuine causal features. It is worth noting that assessing and mitigating spurious correlations in audio remains largely unexplored (Ye et al., 2024).

### 2.4 Bias in Videos

Video understanding datasets exhibit background bias, where models learn to rely on scene context and environmental cues rather than the genuine target signal—human motion patterns—when predicting actions. This spurious correlation between background scenes and action categories emerges naturally in video datasets because certain actions are systematically associated with particular environments during data collection (Chung et al., 2022; Li et al., 2023).

Background bias in action recognition has been comprehensively characterized through evaluation of 74 action recognition models trained on the Kinetics-400 dataset (Chung et al., 2022). Systematic evaluation where background and action cues are deliberately mismatched—presenting videos where the background suggests one action but the human motion indicates another—revealed that models predict background classes at 29.5% accuracy compared to only 16.8% accuracy for actions when evaluated on such mismatched videos (Chung et al., 2022). This represents a substantial and pervasive background bias where models rely on scene cues as strongly as, or more strongly than, motion patterns.

Biases emerge from dataset characteristics where certain actions are spatially clustered: tennis occurs predominantly on tennis courts, cooking predominantly in kitchens, and playing basketball predominantly on basketball courts. During training, models encounter a strong correlation between background and action, and exploit this correlation for improved training performance (Chung et al., 2022; Li et al., 2023). Model design choices significantly influence background bias: models trained on fewer frames per video exhibit stronger background bias, suggesting that temporal information helps mitigate background reliance (Chung et al., 2022).

## 2.5 Related Work

### 2.5.1 Bias Mitigation

Here, we group the relevant bias mitigation approaches based on *how the bias signal is acquired.*

**Direct use of bias labels.** This category assumes that bias attribute annotations are available at training time and employs them directly in the training objective. Several approaches belongs to this category. Group Distributionally Robust Optimization (GroupDRO) (Sagawa et al., 2019) partitions the training set into groups defined by the target and bias attribute classes and minimizes the worst-case per-group loss. Domain Independent (DI) (Wang et al., 2020) routes each input through one of the domain-specific classification heads selected by its bias label. Entangling and Disentangling (EnD) (Tartaglione et al., 2021) uses bias labels to decide which pairs of samples should be pushed together or pulled apart. Bias Balance (BB) (Hong & Yang, 2021) uses the bias labels to compute log-priors that are used to mitigate the biases in the logit space. Idrissi et al. (2022) have further shown that simple group-conditional resampling matches the performance of more complex objectives on standard benchmarks.

**Auxiliary models trained with bias-label supervision.** A second line of work uses bias labels indirectly, i.e., a separate *bias-capturing* network is trained to predict the bias attribute from the input, and the features or outputs of that network are then used in the main training pipeline. BAdd (Sarridis et al., 2025b) adds the bias-capturing features to the main features before the classification head, discouraging the main model from re-encoding bias information. FLAC (Sarridis et al., 2024) minimizes the mutual information between the main and bias-capturing representations. MAVias (Sarridis et al., 2025a) couples a foundation-model-based bias discovery stage with a regularizer for main and bias-capturing logits. ReBias (Bahng et al., 2020) and SoftCon (Hong & Yang, 2021) also rely on auxiliary models that are designed to encode the predefined bias (e.g., color classifiers). The shared property of this category is that the bias type or label is used only as a supervision signal for the auxiliary network, and the signal that ultimately reaches the main model is a continuous, input-conditional feature vector rather than a discrete categorical index. Furthermore, unlike methods that directly incorporate bias labels into the training objective, these approaches are not restricted to the annotations of the main dataset. Instead, auxiliary models can be trained on external data (e.g., a gender-labeled dataset for training a gender classifier) or directly leverage pretrained models, such as the foundation models used in MAVias.

**Auxiliary models without bias-label supervision.** When bias information is not available, an auxiliary model can still be trained, but it must derive its own notion of which samples are bias-conflicting. Learning from Failure (LfF) (Nam et al., 2020) trains an auxiliary classifier with a generalized cross-entropy loss to amplify the bias, and uses the disagreement between this biased model and the main model to up-weight bias-conflicting samples. DebiAN (Li et al., 2022) alternates training of a main model and an auxiliary that identifies subgroups violating equal opportunity. Bias Ensemble (BE) (Lee et al., 2023) extends this idea with multiple biased auxiliary models whose ensemble loss is used to identify bias-conflicting samples more reliably. Disen (Lee et al., 2021) employs a pair of models to perform a feature-level data augmentation technique to synthesize diverse bias-conflicting samples.

**Pseudo-labels through the main model's training dynamics.** A third option for operating without bias labels is to derive pseudo-labels for bias-conflicting samples from the main model itself. Just Train Twice (JTT) (Liu et al., 2021) trains an initial ERM model for a few epochs, treats its misclassified examples as a bias-conflicting set, and upsamples them. Several other methods in this family use representation-space rather than loss-space signals to discover hidden subpopulations: GEORGE (Sohoni et al., 2020) clusters the penultimate features of an ERM model to discover hidden subclasses and BPA (Seo et al., 2022b) clusters representations into pseudo-attributes and reweights samples. Recent work, such as Gradient Extrapolation for Debiased Representation Learning (GERNE) (Asaad et al., 2025) and self-supervised approaches such as Sebra (Adarsh et al., 2025) extend this line.

**No explicit bias signal.** Finally, to the best of our knowledge, there is only one method, namely Spectral Decoupling (SD) (Pezeshki et al., 2021), that does not use any bias signal at all and instead applies a generic regularizer expected to mitigate biases implicitly.

### 2.5.2 Cross-modality Evaluations

Works such as G-AUDIT (Drenkow et al., 2025) have demonstrated the importance of systematic auditing for detecting dataset biases across modalities, including medical imaging, electronic health records, and tabular data. Similarly, research on the transferability of bias mitigation effects has explored whether debiasing techniques applied upstream can generalize across different downstream tasks and domains, revealing that bias mitigation benefits can transfer across text classification, coreference resolution, and other NLP tasks (Jin et al., 2021).

Although the individual bias mitigation methods described above have been primarily evaluated on natural image benchmarks, several of them include evaluations beyond natural images, though always restricted to a single additional modality. Regarding text modality, Sagawa et al. (2019) evaluates GroupDRO on MultiNLI (Williams et al., 2018). Liu et al. (2021) extend JTT's evaluation to MultiNLI and CivilComments-WILDS (Koh et al., 2021), where the spurious attribute is the mention of particular demographic terms in online comments. Subsequent works also consider these benchmarks, namely Idrissi et al. (2022), Kirichenko et al. (2023), and Izmailov et al. (2022). The more recent methods GERNE (Asaad et al., 2025) and Sebra (Adarsh et al., 2025), similarly, combine image benchmarks with a single NLP dataset (CivilComments and MultiNLI, respectively). Regarding medical imaging, EnD (Tartaglione et al., 2021) reports experiments on COVID-19 chest X-ray classification, and GEORGE (Sohoni et al., 2020) is evaluated on the ISIC skin-lesion dataset (Codella et al., 2019). Regarding videos, ReBias (Bahng et al., 2020) is the only method evaluated on video action recognition using the Kinetics (Kay et al., 2017) dataset for training and Mimetics (Weinzaepfel & Rogez, 2021) as a test set for static-scene shortcut removal in action recognition. Notably, none of the existing approaches has been evaluated on audio benchmarks. Overall, prior work is limited in scope, as no study evaluates more than two modalities simultaneously, and there is no unified benchmark assessing a broad range of bias mitigation methods across natural images, text, audio, medical imaging, and video. This work addresses this gap by providing the first large-scale, unified evaluation of 14 bias mitigation methods across multiple modalities, offering empirical insights into their generalization capabilities.

## 3 Methodologies

### 3.1 Notation

Here, we introduce the notation used throughout the paper to ensure consistency across all methods. Let $f(\boldsymbol{x}; \boldsymbol{\theta})$ denote a neural network parametrized by $\boldsymbol{\theta}$, mapping an input $\boldsymbol{x} \in \mathbb{X}$ to a prediction $\hat{y} \in \mathbb{Y}$. The training dataset is denoted as

$$\mathbb{D} = \{(\boldsymbol{x}^{(i)}, y^{(i)}, \alpha^{(i)})\}_{i=1}^{N},$$

with $N$ training examples, where $y^{(i)} \in \mathbb{Y}$ is the target label and $\alpha^{(i)} \in \mathbb{A}$ is a bias attribute.

In this setting, the $i$-th sample is called bias-aligned if its target label $y^{(i)}$ and bias attribute $a^{(i)}$ follow the predominant spurious correlation in the training data, while the $i$-th sample is called bias-conflicting if $y^{(i)}$ and $a^{(i)}$ break this correlation and the attribute value contradicts the dominant bias pattern for that class.

The learned feature representation is $\boldsymbol{z}^{(i)} = h(\boldsymbol{x}^{(i)}; \boldsymbol{\theta})$, $h : \mathbb{X} \to \mathbb{R}^d$, where $h$ denotes a feature extractor and the full model decomposes as $f(\boldsymbol{x}; \boldsymbol{\theta}) = g(h(\boldsymbol{x}; \boldsymbol{\theta}); \boldsymbol{\theta})$, where $g : \mathbb{R}^d \to \mathbb{Y}$ is the classification head. We denote the logits by $\boldsymbol{l}(\boldsymbol{x}^{(i)}; \boldsymbol{\theta}) \in \mathbb{R}^C$, where $C$ is the number of target classes in $\mathbb{Y}$. The vanilla Empirical Risk Minimization (ERM) objective is

$$\min_{\boldsymbol{\theta}} \frac{1}{N} \sum_{i=1}^{N} \mathcal{L}\big(f(\boldsymbol{x}^{(i)}; \boldsymbol{\theta}), y^{(i)}\big),$$

where $\mathcal{L}$ is typically the cross-entropy loss.

For methods requiring group structure, we define a group $g = (y, \alpha) \in \mathbb{G}$, $\mathbb{G} = \mathbb{Y} \times \mathbb{A}$, and the subset of $\mathbb{D}$ belonging to group $g$ as $\mathbb{D}_g = \{(\boldsymbol{x}^{(i)}, y^{(i)}, \alpha^{(i)}) \in \mathbb{D} : y^{(i)} = y, \ \alpha^{(i)} = \alpha\}$, with cardinality $N_g = |\mathbb{D}_g|$.

Table 1 summarizes the notation that is common to all the presented methodologies.

Table 1: Essential notation.

| Symbol | Description |
|---|---|
| $\mathbb{X}$ | Set of input samples |
| $\mathbb{Y}$ | Set of target classes |
| $\mathbb{A}$ | Set of bias attribute classes |
| $\mathbb{D}$ | Training dataset |
| $\mathbb{G}$ | Set of groups ($\mathbb{Y} \times \mathbb{A}$) |
| $\mathbb{T}$ | Set of tags |
| $\mathbb{J}$ | Set of misclassified samples |
| $N$ | Number of samples |
| $C$ | Number of target classes |
| $\boldsymbol{x} \in \mathbb{X}$ | Input sample |
| $y \in \mathbb{Y}$ | Target label |
| $\hat{y} \in \mathbb{Y}$ | Predicted label |
| $\alpha \in \mathbb{A}$ | Bias attribute label |
| $g = (y, \alpha) \in \mathbb{G}$ | Group defined by label–attribute pair |
| $\lambda$ | Method specific hyperparameter |
| $w$ | Weight |
| $\mathcal{L}(\cdot)$ | Loss function |
| $I(\cdot, \cdot)$ | Mutual Information |
| $\sigma(\cdot)$ | Sigmoid function |
| $p(\cdot)$ | Probability |
| $p(\cdot|\cdot)$ | Conditioned probability |
| $\boldsymbol{l}$ | Logits |
| $\boldsymbol{Z}$ | Features matrix |
| $\boldsymbol{G}$ | Gram matrix |
| $\boldsymbol{\theta}$ | Trainable parameters of the main model |
| $\boldsymbol{\psi}$ | Trainable parameters of a method-specific auxiliary model |
| $d$ | Features dimension |
| $h(\cdot)$ | Feature extractor |
| $g(\cdot)$ | Classification head |
| $f(\cdot)$ | Full model (backbone & classification head) |

### 3.2 Bias Mitigation Methods

This study encompasses several approaches of both categories, featuring the following methods: GroupDro (Sagawa et al., 2019), DI (Wang et al., 2020), EnD (Tartaglione et al., 2021), BB (Hong & Yang, 2021), BAdd (Sarridis et al., 2025b), LfF (Nam et al., 2020), SD (Pezeshki et al., 2021), JTT (Liu et al., 2021), Debian (Li et al., 2022), FLAC (Sarridis et al., 2024), MAVias (Sarridis et al., 2025a), BE (Lee et al., 2023), BPA (Seo et al., 2022b), and GEORGE (Sohoni et al., 2020). The selected set covers a wide range of *categories* of bias mitigation strategies w.r.t. two axes. As discussed in Section 2.5.1, the first one concerns how the bias signal is acquired, and involves five categories: *Direct use of bias labels*, *Auxiliary models trained with bias-label supervision*, *Auxiliary models without bias-label supervision*, *Pseudo-labels through the main model's training dynamics*, and *No explicit bias signal*. The second axis concerns the algorithmic intervention and involves: *Loss reweighting* (per-sample or per-group weights on the loss), *Dataset resampling* (physically changing which samples appear in training batches), *Logit-space intervention* (additive corrections or regularizers applied to the logits), *Representation space regularizer* (loss terms applied to the feature representations), *Bias injection* (intentionally injecting a bias-capturing signal into the main forward pass), and *Architectural*

*separation* (distinct classification heads per bias group). The key characteristics of these methods are reported in Table 2. Below, we briefly present the considered approaches.

Table 2: Summary of the integrated bias mitigation methods. *Signal* denotes how the bias signal is acquired and *Intervention* denotes the algorithmic intervention applied.

| Method | Signal | Intervention | Summary |
|---|---|---|---|
| GroupDRO | Direct bias labels access | Loss reweighting | Minimizes worst-case loss across predefined groups. |
| DI | Direct bias labels access | Architectural separation | Uses domain-specific classification heads for domain invariance. |
| EnD | Direct bias labels access | Representation regularizer | Disentangles bias representations and entangles class representations. |
| BB | Direct bias labels access | Logit-space intervention | Balances bias in the logit space using prior bias information. |
| BAdd | Auxiliary w/ bias labels | Bias injection | Adds bias-capturing features to training to discourage their use. |
| FLAC | Auxiliary w/ bias labels | Representation regularizer | Minimizes mutual information between representations and bias-capturing features. |
| MAVias | Auxiliary w/ bias labels | Bias injection | Infers and mitigates visual biases using foundation models and regularization. |
| LfF | Auxiliary w/o bias labels | Loss reweighting | Reweights samples based on bias-conflicting predictions from an auxiliary model. |
| Debian | Auxiliary w/o bias labels | Loss reweighting | Alternates training of main and auxiliary models for debiasing. |
| BE | Auxiliary w/o bias labels | Loss reweighting | Uses an ensemble of biased auxiliary models to identify bias-conflicting samples. |
| JTT | Pseudo-labels through main model | Dataset resampling | Reweights misclassified samples, assuming they are bias-conflicting. |
| GEORGE | Pseudo-labels through main model | Loss reweighting | Clusters ERM features to discover hidden subclasses. |
| BPA | Pseudo-labels through main model | Loss reweighting | Clusters representations into pseudo-attributes and reweights. |
| SD | No explicit bias signal | Logit-space intervention | Regularizes network logits for spectral decoupling and bias robustness. |

**Group Distributionally Robust Optimization (GroupDRO).** GroupDRO (Sagawa et al., 2019) addresses bias by minimizing the worst-case loss across predefined groups $\mathbb{G}$. It maintains group weights $\{w_g\}_{g \in \mathbb{G}}$ updated during training. At iteration $t$, the weights are updated as

$$w_g^{(t+1)} = w_g^{(t)} \exp\left(\lambda_{GroupDro} \mathcal{L}_g^{(t)}\right),$$

where $\lambda_{GroupDro} > 0$ is a step size and $\mathcal{L}_g^{(t)}$ is the loss for group $g$ at iteration $t$. Weights are normalized:

$$w_g^{(t+1)} \leftarrow \frac{w_g^{(t+1)}}{\sum_{g' \in \mathbb{G}} w_{g'}^{(t+1)}}.$$

The model minimizes the weighted average loss:

$$\min_{\boldsymbol{\theta}} \sum_{g \in \mathbb{G}} w_g^{(t)} \cdot \frac{1}{N_g} \sum_{(\boldsymbol{x}^{(i)}, y^{(i)}, \alpha^{(i)}) \in \mathbb{D}_g} \mathcal{L}\left(f(\boldsymbol{x}^{(i)}; \boldsymbol{\theta}), y^{(i)}\right).$$

**Domain Independent (DI).** DI (Wang et al., 2020) aims to mitigate bias by learning representations invariant across different domains. Unlike traditional single-head architectures, DI employs multiple domain-specific classification heads $\{g^{(\alpha)}\}_{\alpha \in \mathbb{A}}$, where each head $g^{(\alpha)} : \mathbb{R}^d \rightarrow \mathbb{Y}$ corresponds to a distinct domain.

Let $h : \mathbb{X} \rightarrow \mathbb{R}^d$ be the shared feature extractor. For each input $(\boldsymbol{x}^{(i)}, y^{(i)}, \alpha^{(i)})$ belonging to domain $\alpha^{(i)}$, the model selects the classification head associated with this domain: $g^{(\alpha^{(i)})}\left(h(\boldsymbol{x}^{(i)}; \boldsymbol{\theta})\right)$.

The training objective minimizes the domain-specific loss:

$$\min_{\boldsymbol{\theta}} \frac{1}{N} \sum_{i=1}^{N} \mathcal{L}\left(g^{(\alpha^{(i)})}\left(h(\boldsymbol{x}^{(i)}; \boldsymbol{\theta})\right), y^{(i)}\right).$$

**Entangling and Disentangling (EnD).** EnD (Tartaglione et al., 2021) explicitly manipulates the feature space to reduce spurious correlations by working on the Gram matrix (inner products between normalized features). The method encourages features with the same bias attribute but different target classes to be orthogonal (disentangled), while encouraging features with the same target class but different bias attributes to be parallel (entangled). Each feature vector is normalized as:

$$\tilde{\boldsymbol{z}}^{(i)} = \frac{\boldsymbol{z}^{(i)}}{\|\boldsymbol{z}^{(i)}\|},$$

and the Gram matrix $\boldsymbol{G} \in \mathbb{R}^{M \times M}$ is computed as:

$$\boldsymbol{G} = \tilde{\boldsymbol{Z}}^{\top} \tilde{\boldsymbol{Z}},$$

where each entry $G_{i,j} = (\tilde{\boldsymbol{z}}^{(i)})^{\top} \tilde{\boldsymbol{z}}^{(j)} \in [-1, +1]$ represents the cosine similarity between normalized feature pairs.

The overall EnD regularization combines disentangling and entangling terms. The disentanglement term, $\mathcal{L}_{\perp}(\boldsymbol{G}, \mathbb{D}, \boldsymbol{\theta})$, encourages samples with the same bias attribute but different target classes to have orthogonal representations by minimizing their correlations, while the entanglement term, $\mathcal{L}_{\parallel}(\boldsymbol{G}, \mathbb{D}, \boldsymbol{\theta})$, encourages samples with the same target class but different bias attributes to have parallel (aligned) representations by maximizing their inner products. Please refer to Tartaglione et al. (2021) for details on the implementation of $\mathcal{L}_{\perp}$ and $\mathcal{L}_{\parallel}$).

The overall training objective combines the task-specific loss with EnD regularization:

$$\min_{\boldsymbol{\theta}} \frac{1}{N} \sum_{i=1}^{N} \mathcal{L}\big(f(\boldsymbol{x}^{(i)}; \boldsymbol{\theta}), y^{(i)}\big) + \lambda_{\text{EnD},1} \mathcal{L}_{\perp}(\boldsymbol{G}, \mathbb{D}, \boldsymbol{\theta}) + \lambda_{\text{EnD},2} \mathcal{L}_{\parallel}(\boldsymbol{G}, \mathbb{D}, \boldsymbol{\theta}),$$

where regularization terms are denoted as $\lambda_{\text{EnD},1}$ and $\lambda_{\text{EnD},2}$.

**Bias Balance (BB).**  BB(Hong & Yang, 2021) addresses spurious correlations by explicitly correcting for bias in the logit space. The method assumes access to prior knowledge about the bias distribution, specifically the conditional probability $p(\alpha \mid y)$ of observing bias attribute $\alpha$ given target class $y$ in the training data. For a given sample with bias attribute $\alpha$, BB constructs a bias prior vector $\boldsymbol{p}^{(\alpha)} \in \mathbb{R}^{C}$ where each entry corresponds to a target class:

$$[\boldsymbol{p}^{(\alpha)}]_y = \log p(\alpha \mid y), \quad \forall y \in \mathbb{Y}.$$

The prior $p(\alpha \mid y)$ is estimated empirically from the training set:

$$p(\alpha \mid y) = \frac{N_{y,\alpha}}{N_y},$$

where $N_{y,\alpha}$ is the number of training samples with target label $y$ and bias attribute $\alpha$, and $N_y$ is the total number of samples with target $y$. Then, the training objective incorporates this bias-balancing mechanism:

$$\min_{\boldsymbol{\theta}} \frac{1}{N} \sum_{i=1}^{N} \mathcal{L}\big(f(\boldsymbol{x}^{(i)}; \boldsymbol{\theta}) - \boldsymbol{p}(\alpha^{(i)}), y^{(i)}\big),$$

where $\boldsymbol{p}^{(\alpha^{(i)})}$ is the bias prior vector corresponding to the bias attribute $\alpha^{(i)}$ of sample $i$. By subtracting the log-prior $\log p(\alpha \mid y)$ from the logits, BB effectively removes the statistical bias encoded in the training distribution, forcing the model to learn decision boundaries that are independent of the spurious correlation structure between target labels and bias attributes.

**Bias Addition (BAdd).**  BAdd (Sarridis et al., 2025b) explicitly adds bias-capturing features to the training procedure to discourage the main model from learning such features. First, the method trains an auxiliary bias-capturing model $f(\boldsymbol{x}; \boldsymbol{\psi})$ to predict bias attribute $\alpha$:

$$\min_{\boldsymbol{\psi}} \frac{1}{N} \sum_{i=1}^{N} \mathcal{L}\big(f(\boldsymbol{x}^{(i)}; \boldsymbol{\psi}), \alpha^{(i)}\big).$$

Let $h(\boldsymbol{x}^{(i)}; \boldsymbol{\psi}) \in \mathbb{R}^d$ be bias-capturing features and $h(\boldsymbol{x}^{(i)}; \boldsymbol{\theta}) \in \mathbb{R}^d$ be the main model features. Then the training objective is:

$$\min_{\boldsymbol{\theta}} \frac{1}{N} \sum_{i=1}^{N} \mathcal{L}\big(g(h(\boldsymbol{x}^{(i)}; \boldsymbol{\theta}) + h(\boldsymbol{x}^{(i)}; \boldsymbol{\psi}); \boldsymbol{\theta}), y^{(i)}\big).$$

The underlying idea of BAdd is to inject $h(\boldsymbol{x}^{(i)}; \boldsymbol{\psi})$ into the training procedure, discouraging the main model from re-encoding bias information, as doing so would not further minimize training loss. Thus, $f(\boldsymbol{x}; \boldsymbol{\theta})$ learns different features that are expected to be more relevant to the defining characteristics of the target class.

**Fairness Aware Representation Learning (FLAC).** FLAC (Sarridis et al., 2024) focuses on learning fair representations by minimizing the dependence between features and sensitive attributes. The objective is to minimize the mutual information between representations and sensitive attributes, defined as:

$$\min_{\boldsymbol{\theta}} \mathcal{L}\big(f(\boldsymbol{x};\boldsymbol{\theta}), y\big) + \lambda_{\text{FLAC}} I(h(\boldsymbol{x};\boldsymbol{\theta}), h(\boldsymbol{x};\boldsymbol{\psi})),$$

where $\lambda_{\text{FLAC}}$ is a hyperparameter, $I(h(\boldsymbol{x};\boldsymbol{\theta}), h(\boldsymbol{x};\boldsymbol{\psi}))$ is the mutual information between representations and sensitive attributes. FLAC employs a bias-capturing model, $f(\boldsymbol{x};\boldsymbol{\psi})$, trained to derive features $h(\boldsymbol{x};\boldsymbol{\psi})$. In scenarios where training a dedicated bias-capturing model is impractical (e.g., unknown biases), the biased vanilla model can be employed, resulting in the FLAC-B variant.

**Mitigate Any Visual Bias (MAVias).** MAVias (Sarridis et al., 2025a) employs a two-stage approach consisting of two decoupled components: (i) a *bias-discovery* stage that leverages foundational models to discover unknown potential biases in an open-set manner, by generating descriptive tags for input images and assessing their relevance to the target class; and (ii) a *bias-mitigation* stage that encodes the identified biases using a vision-language model and incorporates them into training as regularization, discouraging the model from learning spurious correlations. Formally, for each input $\boldsymbol{x}^{(i)}$, the discovery stage first extracts a set of tags $\mathbb{T}_i$ using an image tagging model. The irrelevant subset $\mathbb{T}_i' \subset \mathbb{T}_i$ is selected via a large language model. All tags in $\mathbb{T}_i'$ are embedded together into $h(\boldsymbol{x}^{(i)};\boldsymbol{\psi}) \in \mathbb{R}^d$ using a vision-language encoder and a projection layer. In the mitigation stage, the main model backbone produces features $h(\boldsymbol{x}^{(i)};\boldsymbol{\theta})$ and logits $\boldsymbol{l}_{\boldsymbol{\theta}}(\boldsymbol{x}^{(i)})$, while the bias-capturing features $h(\boldsymbol{x}^{(i)};\boldsymbol{\psi})$ are passed through the same classification head $g$ to obtain $\boldsymbol{l}_{\boldsymbol{\psi},\boldsymbol{\theta}}(\boldsymbol{x}^{(i)})$. The final logits for loss computation are their sum. The overall minimization objective is calculated as:

$$\min_{\boldsymbol{\theta},\boldsymbol{\psi}} \mathcal{L}\big(f(\boldsymbol{x}^{(i)};\boldsymbol{\theta},\boldsymbol{\psi}), y^{(i)}\big) + \lambda_{\text{MAVias},1} \mathcal{L}_{\text{align}}\big(\boldsymbol{l}(\boldsymbol{x}^{(i)};\boldsymbol{\theta}), \boldsymbol{l}(\boldsymbol{x}^{(i)};\boldsymbol{\theta},\boldsymbol{\psi}), \lambda_{\text{MAVias},2}\big),$$

where $\lambda_{\text{MAVias},1}$ and $\lambda_{\text{MAVias},2}$ are hyperparameters and the following regularization term is used:

$$\mathcal{L}_{\text{align}} = \frac{1}{2} \big\| \|\boldsymbol{l}(\boldsymbol{x};\boldsymbol{\theta})\| - \lambda_{\text{MAVias},2}\|\boldsymbol{l}(\boldsymbol{x};\boldsymbol{\theta},\boldsymbol{\psi})\| \big\|^2.$$

Our study targets the mitigation module exclusively. The bias discovery module is not transferable to the present setting for two reasons. First, it is built on top of foundation models that do not apply to the considered modalities. Second, the benchmarks we consider fall under *closed-set* bias settings with a single spurious correlation, rendering the *open-set* discovery component unnecessary. Since MAVias, BAdd, and FLAC share the same requirement for extracting a bias signal (i.e., a model whose features encode the spurious attribute), the bias-capturing component can be treated as a plug-and-play module. Therefore, for consistency, we adopt the same auxiliary bias-capturing model used by BAdd and FLAC in the MAVias experiments. Note that a side benefit of this choice is that, since BAdd, FLAC, and MAVias share the same bias signal, the performance differences among them isolate the contribution of their respective mitigation mechanisms.

**Learning from Failure (LfF).** LfF (Nam et al., 2020) employs a dual-model training paradigm that simultaneously trains the main model $f(\boldsymbol{x};\boldsymbol{\theta})$ and an auxiliary model $f(\boldsymbol{x};\boldsymbol{\psi})$. The key insight is that the biased model is intentionally trained to amplify bias by focusing on "easy" (bias-aligned) samples, while the debiased model focuses on samples that the biased model struggles with (bias-conflicting samples).

The biased model $f(\boldsymbol{x};\boldsymbol{\psi})$ is trained using the Generalized Cross Entropy (GCE) loss to amplify its early-stage predictions and make it follow the unintended decision rule based on the bias attribute. Then, for each sample $\boldsymbol{x}^{(i)}$, LfF computes a relative difficulty score that indicates how likely the sample is to be bias-conflicting, where $\mathcal{L}$ denotes the standard cross-entropy loss. The optimization objective is defined as:

$$\min_{\boldsymbol{\theta}} \frac{1}{N} \sum_{i=1}^{N} w_i \, \mathcal{L}\big(f(\boldsymbol{x}^{(i)};\boldsymbol{\theta}), y^{(i)}\big),$$

where $w_i$ denotes the weight assigned to sample $i$.

**Debiasing Alternate Networks (Debian).** Debian (Li et al., 2022) uses an auxiliary model, $f(\boldsymbol{x}; \boldsymbol{\psi})$, that aims to partition training samples for each target class $y$ into two bias groups, $\{\alpha^+, \alpha^-\}$, such that the main model, $f(\boldsymbol{x}; \boldsymbol{\theta})$, violates the Equal Opportunity criterion. In other words, it identifies class-wise splits of the data where the model's True Positive Rate differs significantly across groups. Formally, the output logits are converted to bias group probabilities:

$$p(\boldsymbol{x}^{(i)}) = \sigma\big(\boldsymbol{l}(\boldsymbol{x}^{(i)}; \boldsymbol{\psi})\big) \in [0, 1],$$

where $\sigma(\cdot)$ is the sigmoid function. This probability indicates the likelihood that sample $i$ belongs to the "positive" bias group (higher values mean stronger assignment to the positive group). The decision threshold is 0.5.

For each target class $y$, the average predicted probability w.r.t. the two groups, $\alpha^+$ and $\alpha^-$, is denoted as $P_{\alpha^+}^{(y)}$ and $P_{\alpha^-}^{(y)}$, respectively. $P$ is used to identify the bias-conflicting groups per class. For instance, if $P_{\alpha^-}^{(y)} > P_{\alpha^+}^{(y)}$, then the samples with $(\alpha^+, y)$ are considered as bias-conflicting. Based on this, the weights are calculated as follows:

$$w_i = \begin{cases} 1 + p(\boldsymbol{x}^{(i)}) & \text{if } P_{\alpha^-}^{(y^{(i)})} > P_{\alpha^+}^{(y^{(i)})}, \\ 1 + \big(1 - p(\boldsymbol{x}^{(i)})\big) & \text{if } P_{\alpha^-}^{(y^{(i)})} \leq P_{\alpha^+}^{(y^{(i)})}. \end{cases}$$

Thus, samples that are considered as bias-conflicting with high confidence are up-weighted. Then, the minimization objective is:

$$\min_{\boldsymbol{\theta}} \frac{1}{N} \sum_{i=1}^{N} w_i \, \mathcal{L}\big(f(\boldsymbol{x}^{(i)}; \boldsymbol{\theta}), y^{(i)}\big).$$

The training follows an alternating scheme that allows $f(\boldsymbol{x}; \boldsymbol{\psi})$ to continuously identify new biases as $f(\boldsymbol{x}; \boldsymbol{\theta})$ evolves, and $f(\boldsymbol{x}; \boldsymbol{\theta})$ to progressively mitigate these identified biases.

**Bias Ensemble (BE).** BE (Lee et al., 2023) suggests an ensemble-based data-selection procedure that filters bias-conflicting samples out of the training set used for $f(\boldsymbol{x}; \boldsymbol{\psi})$. In a first stage, $M$ auxiliary biased models $\{f(\boldsymbol{x}; \boldsymbol{\psi}_m)\}_{m=1}^{M}$ are pretrained with the GCE loss from different random initializations. For each sample $\boldsymbol{x}^{(i)}$ and each model $m$, the ground-truth softmax probability

$$p_m^{(i)} = \big[\sigma(f(\boldsymbol{x}^{(i)}; \boldsymbol{\psi}_m))\big]_{y^{(i)}}$$

is recorded, and a sample is marked as *bias-aligned* if it exceeds a confidence threshold $\tau$ in at least $\kappa$ of the $M$ models:

$$\mathbb{M} = \Big\{i : \big|\{m : p_m^{(i)} > \tau\}\big| \geq \kappa\Big\}.$$

In the second stage, a fresh biased model $f(\boldsymbol{x}; \boldsymbol{\psi})$ is trained with the standard cross-entropy loss *restricted* to the selected subset $\mathbb{M}$, while the main model $f(\boldsymbol{x}; \boldsymbol{\theta})$ is trained on the full training set with LfF-style reweighting.

**Just Train Twice (JTT).** JTT (Liu et al., 2021) focuses on learning from misclassified examples, as they tend to be bias-conflicting samples. It uses a re-weighting scheme to prioritize these examples during training by modifying the data loaders accordingly. First, it trains an initial model $f(\boldsymbol{x}; \boldsymbol{\theta}_0)$ using standard cross-entropy loss for $\lambda_{\text{JTT},1}$ epochs, and identifies the following error set:

$$\mathbb{J} = \Big\{i : \arg\max_{y \in \mathbb{Y}} f(\boldsymbol{x}^{(i)}; \boldsymbol{\theta}_0) \neq y^{(i)}\Big\}.$$

Then, the new training set is constructed by upsampling the error examples by a factor $\lambda_{\text{JTT},2}$ and keeping one copy of all other samples:

$$\mathbb{D}' = \big\{(\boldsymbol{x}^{(i)}, y^{(i)}) : i \notin \mathbb{J}\big\} \cup \underbrace{\bigcup_{k=1}^{\lambda_{\text{JTT},2}} \big\{(\boldsymbol{x}^{(i)}, y^{(i)}) : i \in \mathbb{J}\big\}}_{\text{upsampled error examples}}.$$

If we denote the re-weighted samples as $\boldsymbol{x}'^{(i)}$ and the corresponding targets as $y'^{(i)}$ with $i \in \{1, \ldots, N'\}$, then the loss is calculated as follows:

$$\min_{\boldsymbol{\theta}} \frac{1}{N'} \sum_{i=1}^{N'} \mathcal{L}\big(f(\boldsymbol{x}'^{(i)}; \boldsymbol{\theta}), y'^{(i)}\big).$$

**GEORGE.** GEORGE (Sohoni et al., 2020) targets the *hidden stratification* problem, in which each observed class $y \in \mathbb{Y}$ contains multiple unlabeled fine-grained subclasses with systematically different error rates. GEORGE proceeds in two stages. First, an ERM model $f(\boldsymbol{x}; \boldsymbol{\psi})$ is trained on the observed superclass labels, and its penultimate features $h(\boldsymbol{x}^{(i)}; \boldsymbol{\psi})$ are collected. Within each superclass $y$, the features are projected to a low-dimensional space via UMAP and then clustered with a Gaussian mixture model using an over-clustering strategy in which the number of clusters $k_y$ is selected automatically via the Silhouette criterion (Rousseeuw, 1987), after which small clusters are filtered. This produces pseudo-subclass assignments $\hat{s}^{(i)} \in \{1, \ldots, k_{y^{(i)}}\}$ and an augmented group structure $\mathbb{G} = \{(y, \hat{s}) : y \in \mathbb{Y}, \hat{s} \in \{1, \ldots, k_y\}\}$. In the second stage, GEORGE optimizes GroupDRO (Sagawa et al., 2019) over these recovered groups.

**Bias Pseudo-Attributes (BPA).** BPA (Seo et al., 2022b) assumes no access to bias annotations and instead recovers *pseudo-attributes* by clustering the feature space of a pretrained baseline model. A base model $f(\boldsymbol{x}; \boldsymbol{\psi})$ is first trained with standard cross-entropy, and its penultimate features $h(\boldsymbol{x}^{(i)}; \boldsymbol{\psi})$ are extracted for every training sample. A clustering function $h_c(\boldsymbol{x}^{(i)}; \boldsymbol{\psi})$ (i.e., $k$-means with $K$ clusters) is then fit to these features and yields a cluster assignment $c^{(i)} = h_c(\boldsymbol{x}^{(i)}; \boldsymbol{\psi}) \in \{1, \ldots, K\}$ for each sample. Let $\mathbb{C}_k = \{i : c^{(i)} = k\}$ and $N_k = |\mathbb{C}_k|$. The resulting clusters act as pseudo-groups that play the same role as the $(y, \alpha)$ groups in GroupDRO but without requiring bias labels. The main model $f(\boldsymbol{x}; \boldsymbol{\theta})$ is then trained with cluster-wise reweighting. At each training iteration, the importance weight of cluster $k$ is updated by:

$$\omega_k \leftarrow (1 - m)\,\omega_k + m \cdot \frac{1}{N_k} \sum_{i \in \mathbb{C}_k} \mathcal{L}\big(f(\boldsymbol{x}^{(i)}; \boldsymbol{\theta}), y^{(i)}\big),$$

where $m$ denotes the momentum. The training objective is

$$\min_{\boldsymbol{\theta}} \sum_{i=1}^{B} w_i\, \mathcal{L}\big(f(\boldsymbol{x}^{(i)}; \boldsymbol{\theta}), y^{(i)}\big).$$

**Spectral Decouple (SD).** SD (Pezeshki et al., 2021) introduces a generic regularization approach that does not require explicit bias identification or auxiliary models. The method is based on the observation that adding a specific regularization term to the network logits leads to spectral decoupling in the learned representations, enhancing robustness against spurious correlations. Specifically, the objective is defined as:

$$\min_{\boldsymbol{\theta}} \mathcal{L}\big(f(\boldsymbol{x}; \boldsymbol{\theta}), y\big) + \lambda_{\mathrm{SD}} \big\| f(\boldsymbol{x}; \boldsymbol{\theta}) \big\|,$$

where $\lambda_{\mathrm{SD}}$ is the regularization weight.

## 4 Evaluation Methodology

### 4.1 Datasets and Evaluation Protocol

As shown in Table 3, we employ a diverse set of 6 datasets spanning text, audio, medical imaging, and video data. Each dataset presents a distinct manifestation of spurious correlations that emerge due to certain domain characteristics. For every dataset, we identify a *target attribute* representing the primary classification task and a *spurious attribute* that exhibits statistical correlation with the target in the training distribution but does not reflect genuine causal relationships. This section provides detailed descriptions of each dataset, including the nature of the data, task formulation, and the sources of spurious correlations.

As detailed in the following subsections, a subset of the considered datasets employs deliberately injected spurious correlations, where the training distribution is subsampled so that the spurious attribute co-occurs

with the target label at a fixed rate. This form of controlled bias injection is a long-standing practice in the vision debiasing literature (Sagawa et al., 2019; Hong & Yang, 2021; Sarridis et al., 2024; 2025b) and characterizes most of the benchmarks on which the compared methods were originally validated (e.g., Waterbirds and UrbanCars). Its main benefit is that it isolates the effect of a single, well-defined spurious correlation at a known level. Therefore, adopting the same protocol in other modalities keeps our evaluation aligned with their original evaluation settings on images. Note that benchmarking on complex, unknown bias scenarios is beyond the scope of this work and constitutes an important direction for future research.

Furthermore, following relevant literature, we adopt Average Accuracy (Avg Acc) and Worst-Group Accuracy (WG Acc) as the primary metrics for evaluating bias mitigation effectiveness (Sagawa et al., 2019; Sarridis et al., 2025a). The only exception applies to benchmarks explicitly designed for bias assessment, which consist exclusively of bias-conflicting (or out-of-distribution) samples; such benchmarks typically report standard accuracy as the evaluation metric (Sarridis et al., 2025c). Groups are defined as combinations of target classes and spurious attributes, with the total number of groups given by $|\mathcal{Y}| \times |\mathcal{A}|$, where $|\cdot|$ denotes cardinality. Avg Acc measures overall model performance aggregated across all groups, while WG Acc ( i.e., the minimum accuracy across any single group) captures the most disadvantaged subgroup.

Table 3: Summary of datasets used for evaluating bias mitigation methods across modalities and domains.

| Dataset | Modality | Target Task | Spurious Attribute |
|---|---|---|---|
| Bias in Bios | Text | Profession | Gender |
| Jigsaw Toxic Comments | Text | Identity hate detection | Identity terms |
| UrbanSounds8K | Audio | Engine idle vs. Siren | Sound salience |
| Speech Accent Archive | Audio | Native vs. Non-native accent | Gender |
| CheXpert + NIH | Medical Imaging | Pneumothorax detection | Dataset source |
| UCF101 + SCUBA | Video | Action recognition | Scene context |

### 4.1.1 Bias in Bios: Occupational Gender Bias

The Bias in Bios (De-Arteaga et al., 2019) dataset addresses spurious correlations between gender and profession in biographies (i.e., text data). The dataset consists of short professional biographies scraped from Common Crawl[1], where each biography describes an individual's occupation and career background. The target classification task is to predict the individual's *profession* out of 28 possible occupational categories (e.g., surgeon, nurse, teacher, software engineer, architect).

The spurious correlation in this dataset emerges from historical and societal patterns of occupational gender imbalance that became embedded in the data collection process. Certain professions are significantly overrepresented by one gender in the training data; for instance, nurses are predominantly female, while surgeons are predominantly male. This reflects real-world historical gender distributions in these occupations, creating strong statistical associations between *gender* (the non-causal spurious attribute) and profession labels that models can exploit during training. Models trained on this data learn to use gender-associated linguistic cues (e.g., gendered pronouns "he" vs. "she", gendered names, or gender-stereotypical descriptive language) as predictive features for profession classification.

### 4.1.2 Jigsaw Toxic Comments: Lexical Bias in Hate Speech Detection

The Jigsaw Toxic Comments (Jigsaw, 2018) dataset contains user comments from online discussion forums, labeled for various forms of toxicity, including threats, obscenity, insults, and identity-based hate speech. In our experiments, we focus specifically on *identity hate* classification as the target task.

The spurious correlation in this dataset (often referred to as lexical bias (Garg et al., 2023)) manifests at the token level, where certain non-offensive identity-related terms (e.g., "muslim", "black", or "jewish") appear disproportionately frequently in comments labeled as hateful. As a consequence, models trained on this dataset learn to associate the presence of certain neutral words with toxicity, leading them to incorrectly

---

[1]https://commoncrawl.org/

classify neutral or positive comments that involve such words as toxic. For example, a comment stating "I'm proud to be muslim" or "Black Lives Matter is an important movement" can be misclassified as identity-hate comments.

The official test set of this dataset does not adequately reveal the extent of lexical bias because it maintains similar distributional characteristics to the training data. To properly evaluate whether models rely on identity terms as spurious shortcuts versus learning genuine markers of hateful content, we construct a new hard-negative test set specifically designed to break the spurious identity-toxicity correlation.

To create this test set, we first perform a statistical analysis of the training data to identify words that appear with significantly elevated frequency in hateful comments but are not inherently offensive. Specifically, we extract the top-300 words that co-occur with the identity hate class within the training data, and then we manually filter this list of words to retain only those terms that are not inherently offensive, resulting in a curated set of the following 33 words: homosexual, homos, lesbians, homosexuals, queer, homosexuality, jew, semite, malaysians, nazi, asian, mexicans, jewish, spanish, white, black, pakistanis, moslem, semites, arabian, nazis, hispanics, indians, turks, mom, moms, males, mums, dads, albino, republican, muslims, and vegans. Then, we leverage a large language model (GPT-4) to generate 10 synthetic test samples for each extracted word, deliberately including them in neutral, non-hateful contexts. We report the generated test set in the Appendix A. We use the accuracy on the original test set and the generated test set to measure overall performance and bias reduction, respectively. It is worth noting that this synthetic data is used *only at the inference stage*, thus any potential artifacts introduced by GPT-4 cannot influence the mitigation process of the compared methods.

### 4.1.3   UrbanSounds8K: Acoustic Salience Bias

The UrbanSounds8K (Salamon et al., 2014) contains 8,732 labeled audio excerpts ($\leq 4$ seconds) of urban sounds from 10 classes: air conditioner, car horn, children playing, dog bark, drilling, engine idling, gunshot, jackhammer, siren, and street music.

In this scenario, we focus on a subset of this dataset where acoustic recording conditions introduce a clear spurious correlation. Specifically, we consider a binary classification task distinguishing between *engine idle* and *siren* sounds, where the spurious attribute is *sound salience* (binary attribute), which is already provided in the data annotations and shows considerably high (low) co-occurrence with the *siren* (*engine idle*) class. Note that apart from class-defining sound properties, also recording circumstances, microphone placement, and environmental factors present during data collection can affect sound salience. Thus, we expect that a robust model can classify both sound types accurately regardless of salience conditions.

### 4.1.4   Speech Accent Archive: Gender-Accent Spurious Correlation

The Speech Accent Archive (Weinberger & Kunath, 2011) dataset consists of speech recordings from speakers with diverse native language backgrounds reading a standardized English passage. Each recording is annotated with speaker demographics (age, gender, native language) and linguistic characteristics. The dataset provides a rich resource for studying accent variation and speech patterns across different language backgrounds.

To investigate whether speaker gender can act as a spurious shortcut in speech classification tasks, we deliberately construct a biased training subset that introduces an intense correlation (90%) between *speaker gender* (the spurious attribute) and *English accent nativeness* (the target attribute). Native speakers are defined as those who learned English as their first language, and non-native speakers as those with other native languages. In particular, we subsample the original dataset to create a training distribution where (90%) of male speakers are native English speakers and (90%) of female speakers are non-native English speakers. This setup follows established practices in the vision domain (Hong & Yang, 2021; Sarridis et al., 2024) for creating controlled spurious correlation scenarios. Note that this injected correlation does not reflect any genuine linguistic relationship (speaker gender does not causally influence language acquisition patterns or accent characteristics), making it an ideal spurious attribute for evaluation purposes. The binary classification task is to predict whether a given recording comes from a native or non-native English speaker.

### 4.1.5 CheXpert + NIH Chest X-ray: Data Source Bias in Medical Images

Medical imaging datasets present unique challenges in terms of biases due to the data acquisition factors. In this set of experiments, we aim to examine whether the dataset-specific characteristics can act as shortcuts to a disease detection model and whether bias mitigation approaches are effective in such scenarios. To this end, we construct a combined chest X-ray dataset by merging samples from two large-scale public datasets, namely CheXpert (Irvin et al., 2019) (collected from Stanford Hospital) and NIH Chest X-ray (Wang et al., 2017) (collected from the NIH Clinical Center). The target classification task is *pneumothorax* detection, i.e., identifying the presence of collapsed lung. We deliberately construct a training distribution where *dataset source* (CheXpert versus NIH) serves as a spurious attribute that is strongly correlated with the pneumothorax label. Specifically, we ensure that 90% of pneumothorax-positive cases originate from the CheXpert dataset, while 90% of negative cases come from the NIH Chest X-ray dataset.

To assess whether bias mitigation methods successfully eliminate reliance on dataset source, we evaluate models on a balanced test.

### 4.1.6 UCF101: Scene-Action Spurious Correlation

The UCF101 dataset (Soomro et al., 2012) is a widely-used benchmark for human action recognition in videos, containing 13,320 video clips across 101 action categories. Videos are collected from YouTube and include diverse human activities ranging from sports (basketball shooting, tennis swing, kayaking) to daily activities (applying makeup, brushing teeth, typing) to musical performances (playing guitar, playing violin, drumming). Each video is approximately 7 seconds long and captured under realistic, unconstrained conditions with varying camera viewpoints, subject appearance, object scales, background complexity, and lighting conditions.

To systematically evaluate scene-action bias in video action recognition, we leverage the SCUBA benchmark (Li et al., 2023), which provides a curated set of samples with action-scene mismatches. Specifically, SCUBA contains videos where actions are performed in atypical background scenes, creating explicit conflicts between the action and scene attributes. This allows us to measure how well models generalize when the spurious action-scene correlation is deliberately broken.

However, the original SCUBA benchmark is designed primarily for domain generalization setups, where no bias-conflicting samples appear in the training set (i.e., 100% co-occurrence between the target class and the spurious attribute). To fairly assess whether considered bias mitigation methods are effective in mitigating spurious scene-action correlations, we require training-time exposure to bias-conflicting examples, i.e., samples where the action and typical background scene do not align. Therefore, we adopt a modified protocol, where we include two samples per action class from the SCUBA benchmark into the training set. At test time, we evaluate on the remaining SCUBA samples that were not included in the training set, where actions continue to be mismatched with their typical scene contexts. Considering that SCUBA involves only bias-conflicting samples, we measure performance using standard classification accuracy, which is the typical evaluation metric for this benchmark and directly reflects whether models can correctly classify actions regardless of background scene context. Additionally, to provide further insights into the generalization properties of bias mitigation methods, we include full experiments using the original SCUBA benchmark without including any SCUBA samples in the training set in the Appendix A.

### 4.2 Implementation Details

For all experiments, we use Adam optimizer with a learning rate of 0.001, a batch size of 64, and a linear scheduler that divides the learning rate by 10 at specified epochs, unless stated otherwise. For both text datasets, we employ the *all-MiniLM-L6-v2* encoder, a compact sentence transformer model that provides effective semantic representations for text classification tasks. For Bias in Bios, we train for 50 epochs and the learning rate decays at epochs 16 and 33. For Jigsaw Toxic Comments, we train for 200 epochs and apply weight decay of 0.01. The learning rate linearly decays at epochs 70 and 140. For the Speech Accent Archive dataset, we use the *wav2vec2-base-960h* as backbone, which provides strong acoustic representations learned from 960 hours of labeled speech data. Training proceeds for 100 epochs and the learning rate decays at epochs 33 and 66. For UrbanSounds8K, we extract the Mel-Frequency Cepstral Coefficients (MFCC) features

as acoustic representations. We train a classification head for 100 epochs with a weight decay of 0.0001. The learning rate decays at epochs 33 and 66. For the medical images, we employ a ResNet18 pretrained on ImageNet as backbone. The model is trained for 20 epochs and the learning rate decays at epochs 6 and 13. For action recognition, we follow other works (Li et al., 2023) and we employ swin-t3d, a 3D spatiotemporal backbone pretrained on the Kinetics-400 dataset, which captures both spatial and temporal patterns that are essential for video understanding. Given the computational demands of video processing, we train for 30 epochs with a smaller batch size of 32. We use the AdamW optimizer with a more conservative learning rate of 0.000025. Weight decay is set to 0.01 and rather than a linear scheduler, we employ a cosine annealing learning rate schedule with 2.5 epochs of warmup. The full hyperparameter selections for each method are reported in the Appendix A. Note that for the MAVias approach, we use only the mitigation component, as its bias detection component is tailored to natural images and thus cannot be employed in the context of this work. For methods requiring a bias-capturing model, we train it on the same dataset, using the spurious attribute as the target. All experiments were conducted using an NVIDIA A100 GPU, except for the video experiments, which were conducted using an NVIDIA H200 GPU. Each experiment was executed with 5 random seeds, except for the video experiments, where we used 3 random seeds.

# 5 Results

We present the results of our empirical evaluation of 14 bias mitigation methods across 6 datasets spanning text, audio, medical imaging, and video modalities. Our analysis addresses the three research questions we posed in the introduction regarding the generalization capabilities of bias mitigation methods for natural images to other domains and modalities.

## 5.1 RQ1: Do bias mitigation methods originally tested for natural image classification reduce bias in other domains/modalities (text, audio, medical imaging, video)?

Table 4: Performance comparison on Jigsaw Toxic Comments and Bias in Bios datasets

| Method | Jigsaw Toxic Comments | | Bias in Bios | |
|---|---|---|---|---|
| | Official Set Acc | Generated Set Acc | Avg Acc | WG Acc |
| Vanilla | $94.6_{\pm 0.1}$ | $73.5_{\pm 0.4}$ | $46.5_{\pm 40.2}$ | $16.9_{\pm 14.7}$ |
| BB | $75.5_{\pm 0.9}$ (-19.1) | $14.0_{\pm 0.8}$ (-59.5) | $66.1_{\pm 0.3}$ (+19.6) | $0.0_{\pm 0.0}$ (-16.9) |
| Debian | $94.8_{\pm 0.1}$ (+0.2) | $81.6_{\pm 0.7}$ (+8.1) | $67.9_{\pm 0.3}$ (+21.4) | $25.7_{\pm 1.6}$ (+8.8) |
| DI | $93.9_{\pm 0.1}$ (-0.7) | $74.2_{\pm 1.3}$ (+0.7) | $69.2_{\pm 0.5}$ (+22.7) | $31.3_{\pm 4.2}$ (+14.4) |
| EnD | $94.9_{\pm 0.2}$ (+0.3) | $81.6_{\pm 0.7}$ (+8.1) | $52.2_{\pm 34.8}$ (+5.7) | $18.7_{\pm 12.6}$ (+1.8) |
| GroupDro | $91.0_{\pm 1.6}$ (-3.6) | $94.0_{\pm 2.7}$ (+20.5) | $72.6_{\pm 0.7}$ (+26.1) | $52.5_{\pm 3.5}$ (+35.6) |
| JTT | $72.5_{\pm 1.2}$ (-22.1) | $80.8_{\pm 1.5}$ (+7.3) | $45.2_{\pm 1.1}$ (-1.3) | $18.9_{\pm 5.9}$ (+2.0) |
| LfF | $94.7_{\pm 0.2}$ (+0.1) | $81.8_{\pm 0.6}$ (+8.3) | $69.4_{\pm 0.2}$ (+22.9) | $27.9_{\pm 4.0}$ (+11.0) |
| SD | $94.7_{\pm 0.1}$ (+0.1) | $72.8_{\pm 0.3}$ (-0.7) | $67.9_{\pm 0.7}$ (+21.4) | $19.1_{\pm 2.9}$ (+2.2) |
| GEORGE | $94.9_{\pm 0.1}$ (+0.3) | $74.3_{\pm 0.9}$ (+0.8) | $35.7_{\pm 0.3}$ (-10.8) | $0.0_{\pm 0.0}$ (-16.9) |
| BE | $94.6_{\pm 0.1}$ (+0.0) | $70.9_{\pm 1.9}$ (-2.6) | $60.2_{\pm 1.5}$ (+13.7) | $19.6_{\pm 7.3}$ (+2.7) |
| BPA | $94.6_{\pm 0.2}$ (+0.0) | $70.3_{\pm 4.0}$ (-3.2) | $72.5_{\pm 0.3}$ (+26.0) | $46.3_{\pm 1.6}$ (+29.4) |
| FLAC | $85.7_{\pm 1.9}$ (-8.9) | $95.8_{\pm 2.5}$ (+22.3) | $68.3_{\pm 0.8}$ (+21.8) | $29.9_{\pm 3.5}$ (+13.0) |
| BAdd | $88.2_{\pm 1.1}$ (-6.4) | $95.5_{\pm 0.9}$ (+22.0) | $70.1_{\pm 0.6}$ (+23.6) | $43.1_{\pm 1.4}$ (+26.2) |
| MAVIAS | $90.1_{\pm 0.7}$ (-4.5) | $94.0_{\pm 1.1}$ (+20.5) | $73.8_{\pm 0.3}$ (+27.3) | $46.8_{\pm 0.6}$ (+29.9) |

For the text modalities, Table 4 reveals clear performance patterns. On the Jigsaw Toxic Comments dataset, most compared methods effectively mitigate biases, as reflected by their accuracy on the generated test set. In particular, FLAC, BAdd, MAVias, and GroupDro achieve the largest gains (>20%) over the vanilla baseline. These improvements often come at the cost of lower accuracy on the official test set, which is expected, as the models rely less on the spurious correlations present in that set. For the Bias in Bios dataset, which targets occupational gender bias, most evaluated methods show improvements in worst-group accuracy, with

GroupDro achieving the most substantial gain of 35.6%. MAVias, BPA, and BAdd also perform strongly, improving worst-group accuracy by 29.9%, 29.4%, and 26.2%, respectively. Across both text datasets, BB demonstrates poor effectiveness, leading to underfitting and consequently degrades both overall performance and bias.

Table 5: Performance comparison on Speech Accent Archive and UrbanSounds8k.

| Method | Speech Accent Archive | | UrbanSounds | |
| | Avg Acc | WG Acc | Avg Acc | WG Acc |
| --- | --- | --- | --- | --- |
| Vanilla | $71.5_{\pm31.7}$ | $64.9_{\pm29.8}$ | $72.9_{\pm1.8}$ | $47.5_{\pm3.4}$ |
| BB | $84.9_{\pm1.0}$ (+13.4) | $80.5_{\pm2.1}$ (+15.6) | $75.3_{\pm4.3}$ (+2.4) | $65.3_{\pm8.5}$ (+17.8) |
| Debian | $84.1_{\pm0.9}$ (+12.6) | $78.9_{\pm2.3}$ (+14.0) | $71.8_{\pm2.4}$ (-1.1) | $51.3_{\pm5.1}$ (+3.8) |
| DI | $84.0_{\pm1.1}$ (+12.5) | $79.7_{\pm1.7}$ (+14.8) | $74.1_{\pm1.7}$ (+1.2) | $60.5_{\pm4.4}$ (+13.0) |
| EnD | $84.6_{\pm1.1}$ (+13.1) | $78.9_{\pm2.2}$ (+14.0) | $72.5_{\pm2.6}$ (-0.4) | $54.0_{\pm9.2}$ (+6.5) |
| GroupDro | $82.8_{\pm0.9}$ (+11.3) | $75.0_{\pm2.2}$ (+10.1) | $67.8_{\pm4.7}$ (-5.1) | $39.7_{\pm6.7}$ (-7.8) |
| JTT | $78.1_{\pm0.4}$ (+6.6) | $72.7_{\pm2.5}$ (+7.8) | $72.5_{\pm1.7}$ (-0.4) | $49.3_{\pm2.8}$ (+1.8) |
| LfF | $85.0_{\pm1.6}$ (+13.5) | $79.6_{\pm2.0}$ (+14.7) | $71.6_{\pm3.0}$ (-1.3) | $53.0_{\pm7.9}$ (+5.5) |
| SD | $85.3_{\pm1.7}$ (+13.8) | $79.4_{\pm3.1}$ (+14.5) | $72.6_{\pm2.1}$ (-0.3) | $44.7_{\pm5.7}$ (-2.8) |
| GEORGE | $86.6_{\pm2.9}$ (+15.1) | $82.6_{\pm1.9}$ (+17.7) | $72.9_{\pm2.2}$ (+0.0) | $58.7_{\pm6.1}$ (+11.2) |
| BE | $82.1_{\pm2.7}$ (+10.6) | $72.4_{\pm6.7}$ (+7.5) | $55.1_{\pm11.5}$ (-17.8) | $14.0_{\pm31.3}$ (-33.5) |
| BPA | $85.7_{\pm2.6}$ (+14.2) | $79.2_{\pm2.1}$ (+14.3) | $71.6_{\pm3.0}$ (-1.3) | $62.3_{\pm1.5}$ (+14.8) |
| FLAC | $84.8_{\pm0.9}$ (+13.3) | $81.6_{\pm1.1}$ (+16.7) | $71.2_{\pm3.9}$ (-1.7) | $60.2_{\pm5.6}$ (+12.7) |
| BAdd | $82.9_{\pm0.2}$ (+11.4) | $78.4_{\pm6.1}$ (+13.5) | $75.9_{\pm2.2}$ (+3.0) | $65.5_{\pm7.3}$ (+18.0) |
| MAVIAS | $86.4_{\pm1.1}$ (+14.9) | $84.7_{\pm0.2}$ (+19.8) | $68.8_{\pm4.0}$ (-4.1) | $62.7_{\pm6.3}$ (+15.2) |

Table 5 demonstrates substantial bias reduction across both datasets in the audio modality (covering both sound and speech domains). On the Speech Accent Archive, all bias mitigation methods significantly outperformed the vanilla baseline in both average and worst-group accuracy, with all methods except JTT and BE achieving gains greater than 10%. MAVias achieved the highest improvement in worst-group accuracy (19.8%), followed by GEORGE (17.7%). For UrbanSound8K, which examines acoustic salience bias, the results were more variable. BAdd and BB achieved the strongest improvements in worst-group accuracy (18% and 17.8%), while BE, GroupDro, and SD exhibited decreased performance, with BE in particular exhibiting notable instability. Additionally, we observe that most methods lead to small drops in average performance. This suggests that, although the models reduced their reliance on spurious correlations, they were unable to leverage other informative features to improve overall accuracy.

For medical imaging, Table 6 shows that bias mitigation on the CheXpert+NIH combined dataset is particularly challenging. While BAdd, BB, and MAVias achieved notable worst-group accuracy improvements of 33.4%, 30.1%, and 24.0% respectively, several methods, including GroupDro and BE, substantially degraded performance. This suggests that the dataset-specific artifacts present in this domain are especially difficult for existing bias mitigation techniques to overcome.

For the video action recognition task, Table 6 shows mixed performance on the UCF101-SCUBA benchmark. MAVias achieved the greatest improvement with a 4.1% accuracy increase, followed by BPA (+3.7%) and SD (+2.5%). However, several methods performed poorly. Most notably, DI reduced accuracy by 25.4%, indicating that using multiple classification heads in scenarios with a large number of classes (e.g., 101) can be detrimental to the training process.

Figure 1 visualizes the ability of the compared methods to reduce biases across the evaluated datasets by summarizing the cases in which each method increased or decreased bias mitigation performance. The figure clearly shows that methods such as MAVias, BAdd, and FLAC consistently yield performance gains across most domains, whereas methods like GroupDro, BB, and JTT exhibit more inconsistent behavior, with performance drops on several datasets.

Table 6: Performance comparison on CheXpert + NIH and UCF101+SCUBA datasets.

| Method | CheXpert + NIH | | UCF101+SCUBA |
|---|---|---|---|
| | Avg Acc | WG Acc | Acc |
| Vanilla | $61.2_{\pm3.4}$ | $31.0_{\pm3.9}$ | $72.0_{\pm0.3}$ |
| BB | $70.8_{\pm3.9}$ (+9.6) | $61.1_{\pm7.3}$ (+30.1) | $69.4_{\pm2.3}$ (-2.6) |
| Debian | $62.1_{\pm2.8}$ (+0.9) | $31.8_{\pm8.2}$ (+0.8) | $70.1_{\pm1.6}$ (-1.9) |
| DI | $69.0_{\pm1.4}$ (+7.8) | $59.3_{\pm4.4}$ (+28.3) | $46.6_{\pm3.7}$ (-25.4) |
| EnD | $61.7_{\pm4.2}$ (+0.5) | $31.5_{\pm2.5}$ (+0.5) | $72.5_{\pm1.1}$ (+0.5) |
| GroupDro | $58.0_{\pm1.9}$ (-3.2) | $12.5_{\pm10.2}$ (-18.5) | $69.0_{\pm2.2}$ (-3.0) |
| JTT | $54.8_{\pm3.5}$ (-6.4) | $20.1_{\pm9.5}$ (-10.9) | $65.4_{\pm1.4}$ (-6.6) |
| LfF | $60.8_{\pm4.4}$ (-0.4) | $25.0_{\pm10.3}$ (-6.0) | $72.6_{\pm1.0}$ (+0.6) |
| SD | $59.0_{\pm4.4}$ (-2.2) | $25.2_{\pm9.6}$ (-5.8) | $74.5_{\pm1.2}$ (+2.5) |
| GEORGE | $59.1_{\pm4.1}$ (-2.1) | $32.3_{\pm17.5}$ (+1.3) | $54.6_{\pm1.6}$ (-17.4) |
| BE | $57.2_{\pm10.1}$ (-4.0) | $15.0_{\pm20.7}$ (-16.0) | $74.0_{\pm0.8}$ (+2.0) |
| BPA | $58.6_{\pm4.0}$ (-2.6) | $39.0_{\pm2.6}$ (+8.0) | $75.7_{\pm0.2}$ (+3.7) |
| FLAC | $52.9_{\pm3.8}$ (-8.3) | $46.3_{\pm5.8}$ (+15.3) | $73.4_{\pm1.2}$ (+1.4) |
| BAdd | $72.1_{\pm2.3}$ (+10.9) | $64.4_{\pm5.3}$ (+33.4) | $73.2_{\pm2.7}$ (+1.2) |
| MAVIAS | $68.7_{\pm1.5}$ (+7.5) | $55.0_{\pm3.8}$ (+24.0) | $76.1_{\pm1.1}$ (+4.1) |

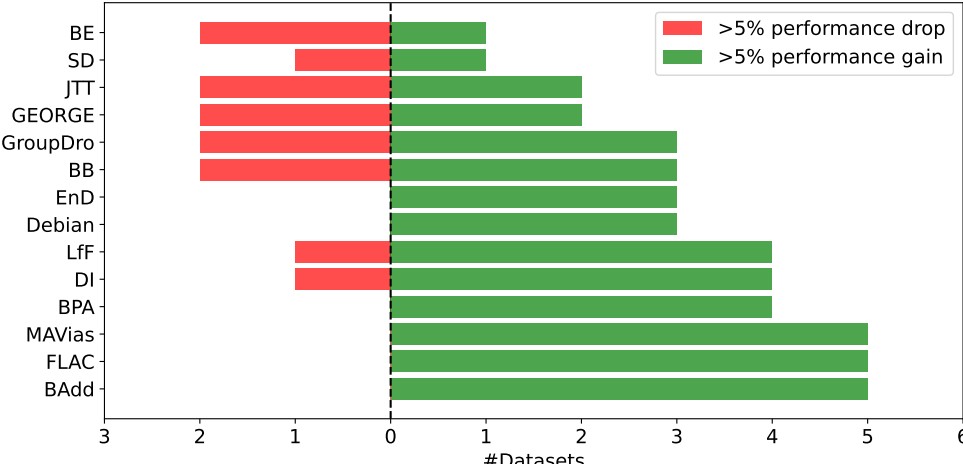

Figure 1: Summary of performance gains and drops for each method across datasets. We define gains/drops as changes in the bias metrics that exceed ±5% relative to the vanilla baselines. Performance corresponds to the fairness metric used for each dataset—generated set accuracy for Jigsaw Toxic Comments, accuracy for UCF101+SCUBA, and worst-group accuracy for all remaining datasets.

To move beyond per-method comparisons and examine which *categories* of bias mitigation strategies generalize across modalities, we further group the 14 evaluated methods w.r.t. two axes defined in Table 2. For each category, we compute the mean worst-group (or task-appropriate) performance of its member methods on every dataset, and report the gap between that mean and the Vanilla baseline.

Table 7 summarizes the results grouped by the bias signal-acquisition mechanism. As one may observe, the *Auxiliary w/ bias labels* category (BAdd, FLAC, MAVias) is the only one to improve over the Vanilla baseline on every one of the six datasets, with an average gain of +17.2% and gains above +20% on Bias in Bios, Jigsaw Toxic Comments, and CheXpert+NIH. In contrast, methods that consume the bias labels *directly* in the training objective (*Direct bias labels access*: GroupDro, DI, EnD, BB) achieve an average gain of only +4.1%, and drop below Vanilla on Jigsaw Toxic Comments (−7.6%) and UCF101+SCUBA (−7.6%). The

Table 7: Per-category performance gap compared to the Vanilla baseline, grouped by the mechanism used to acquire the bias signal. The Avg column is the mean gap across the six datasets. Dataset abbreviations: BiB (Bias in Bios), JTC (Jigsaw Toxic Comments), US (UrbanSounds8k), SAA (Speech Accent Archive), CXN (CheXpert+NIH), UCFS (UCF101+SCUBA).

| Category | BiB | JTC | US | SAA | CXN | UCFS | **Avg** |
|---|---|---|---|---|---|---|---|
| Auxiliary w/ bias labels | +23.0 | +21.6 | +15.3 | +16.7 | +24.2 | +2.2 | **+17.2** |
| Direct bias labels access | +8.7 | -7.6 | +7.4 | +13.6 | +10.1 | -7.6 | +4.1 |
| Pseudo-labels through main model | +4.8 | +1.6 | +9.3 | +13.3 | -0.5 | -6.8 | +3.6 |
| No explicit bias signal | +2.2 | -0.7 | -2.8 | +14.5 | -5.8 | +2.5 | +1.7 |
| Auxiliary w/o bias labels | +7.5 | +4.6 | -8.1 | +12.1 | -7.1 | +0.2 | +1.5 |

Table 8: Per-category performance gap compared to the Vanilla baseline, grouped by the algorithmic intervention mechanism. The Avg column is the mean gap across the six datasets. Dataset abbreviations as in Table 7.

| Category | BiB | JTC | US | SAA | CXN | UCFS | **Avg** |
|---|---|---|---|---|---|---|---|
| Bias injection | +28.0 | +21.2 | +16.6 | +16.7 | +28.7 | +2.7 | **+19.0** |
| Representation regularizer | +7.4 | +15.2 | +9.6 | +15.4 | +7.9 | +1.0 | +9.4 |
| Architectural separation | +14.4 | +0.7 | +13.0 | +14.8 | +28.3 | -25.4 | +7.6 |
| Loss reweighting | +11.8 | +5.3 | -1.0 | +13.1 | -5.1 | -2.7 | +3.6 |
| Dataset resampling | +2.0 | +7.3 | +1.8 | +7.8 | -10.9 | -6.6 | +0.2 |
| Logit-space intervention | -7.4 | -30.1 | +7.5 | +15.1 | +12.2 | -0.1 | -0.5 |

performance gap between these two categories is driven by the fact that methods employing an auxiliary model trained to predict the bias attribute are capable of offering much richer representations than the predefined hard labels. Moreover, as expected, categories that do not involve bias-label supervision (*Auxiliary w/o bias labels*, *Pseudo-labels through main model*, *No explicit bias signal* ) produce smaller average gains (+1.5%, +3.6%, +1.7% respectively), with each of them dropping below Vanilla on at least two datasets.

Table 8 presents the corresponding results grouped by the intervention mechanism. *Bias injection* (BAdd, MAVias) achieves the largest overall gain of any category (+19.0% on average) and never drops below Vanilla; its gains exceed +20% on three of the six datasets. A key factor behind this consistent and stable behavior is that it does not introduce additional optimization objectives; instead, it relies solely on the standard Cross-Entropy loss, thereby reducing the risk of instability. Combined with the ability of those methods to leverage richer bias signals through auxiliary models (as shown in Table 7), makes them both stable and highly effective. The *Representation regularizer* category (EnD, FLAC) is also consistent, having strictly positive gaps on performance, and it averages +9.4% overall. Similar to the previous findings, we observe the benefits of mitigating biases in the representation space. *Loss reweighting* (BPA, GEORGE, BE, Debian, LfF, GroupDro) average only +3.6% performance gains, with only BPA approach showing consistent behavior. *Dataset resampling*, represented solely by JTT, achieves only a marginal average gain (+0.2%) and exhibits highly unstable behavior. This can be attributed to its uniform upsampling of identified bias-conflicting samples, as well as its reliance on the assumption that samples misclassified by the Vanilla model are bias-conflicting, which is strongly dependent on the specific benchmark. *Logit-space intervention* (BB, SD) is the only category with a negative overall gap (−0.5%). *Architectural separation* is populated only by DI and therefore tracks DI's single-method behavior; it is strong on datasets with a small number of classes but suffers a −25.4% drop on UCF101+SCUBA, where it requires learning multiple classification heads with large output dimensionality.

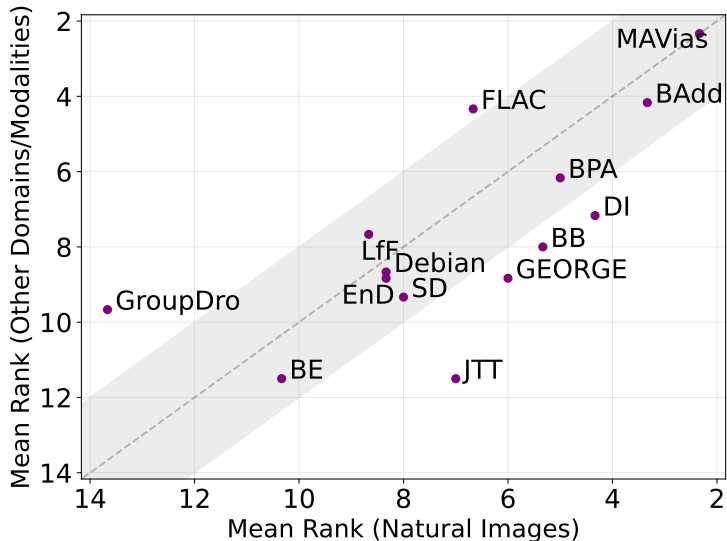

Figure 2: Rankings comparison: Natural image data vs other domains/modalities.

## 5.2 RQ2: How do different methods rank, and are method rankings maintained across different domains/modalities compared to natural image benchmarks?

To comprehensively evaluate the ranking of the methods, we also incorporate results from recent work benchmarking bias mitigation techniques on natural image datasets (Sarridis et al., 2025c). Specifically, we use the reported worst-group accuracy for the same 14 methods evaluated in our study on CelebA, Waterbirds, and UrbanCars.[2]

Figure 2 compares the mean rankings of the methods when evaluated on natural images versus other domains/modalities. As shown, most methods maintain relatively stable rankings across evaluation contexts, typically within approximately $\pm 2$ rank positions, with MAVias and BAdd consistently occupying the top two positions. In contrast, FLAC and GroupDro tend to perform worse on natural images, whereas JTT, GEORGE, BB, and DI achieve better rankings in that setting.

Figure 3 provides a more detailed view of the mean rankings by presenting separate ranking visualizations for natural images (panel a), other domains/modalities (panel b), and overall performance (panel c). The figure also depicts ranking variability, estimated via bootstrapping with 10,000 simulations. Specifically, the variability is obtained by resampling the datasets with replacement and recomputing the rankings for each bootstrap sample. Notably, MAVias, BAdd, and FLAC occupy the top three positions in the overall ranking, with MAVias exhibiting the highest consistency.

## 5.3 Statistical Analysis

To further support the previous findings, we conducted a statistical analysis to assess which bias mitigation methods show statistically significant improvements in bias reduction across diverse modalities. After conducting a Friedman test, which yielded a $p$-value of $1.5 \times 10^{-5}$, Table 9 presents the results of the Nemenyi post-hoc pairwise comparisons across all datasets (including the three natural image datasets), providing statistical evidence regarding the effectiveness of each method. The analysis indicates that only a subset of methods achieves statistically significant improvements over the vanilla baseline. Specifically, MAVias exhibits the strongest statistical evidence for effectiveness, achieving a $p$-value of 0.00013 and an average rank of 2.33. This highly significant result ($p < 0.001$) provides compelling support that MAVias consistently reduces bias across diverse modalities and domains. BAdd obtains the second-strongest statistical evidence,

---

[2]For the additional methods GEORGE, BPA, and BE, we run the experiments on natural images using their open-source official implementations.

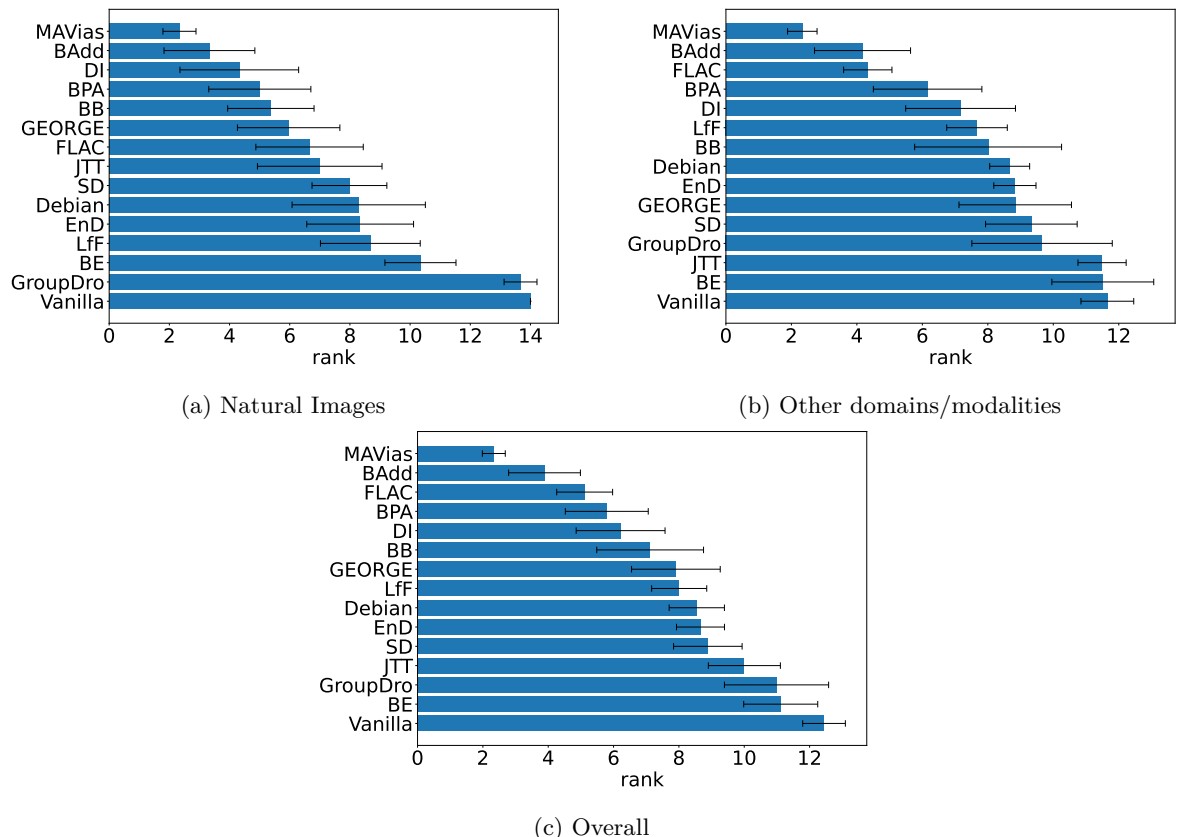

(a) Natural Images

(b) Other domains/modalities

(c) Overall

Figure 3: Method rankings (lower is better).

with a $p$-value of 0.00359 ($p < 0.01$) and an average rank of 3.89. FLAC also shows statistical significance, with a $p$-value of 0.03773 ($p < 0.05$) and an average rank of 5.11.

It is important to note that the Nemenyi post-hoc test is highly conservative, and detecting statistically significant differences is challenging when the number of samples (i.e., datasets) is relatively small (Demšar, 2006). Consequently, observing $p < 0.05$ is a strong indicator of robustness, while the absence of significance should not be interpreted as evidence that a method fails to reduce bias. In addition to the statistical tests, the average rankings presented in Figure 3 provide valuable insights for understanding the effectiveness relationships between the compared methods.

## 6 Conclusion

This work addresses an important research gap in machine learning bias research by conducting the first comprehensive empirical investigation of whether bias mitigation methods originally designed for natural images can generalize to other domains and modalities, evaluating 14 methods across 7 diverse datasets spanning text, audio, medical imaging, and video. Our findings demonstrate that most bias mitigation approaches reduce bias across multiple modalities, with methods such as MAVias, BAdd, and FLAC consistently achieving substantial improvements in worst-group accuracy exceeding 20% in several cases. Regarding method stability, most approaches maintain relatively consistent rankings across natural images and other domains, with MAVias and BAdd consistently occupying top positions, though notable exceptions like FLAC and GroupDro exhibit domain-specific ranking variations. Statistical significance testing using the Friedman test reveals that only a subset of methods achieve statistically significant improvements when aggregated across all datasets, with MAVias (p < 0.001, rank 2.33) and BAdd (p < 0.01, rank 3.22)

Table 9: Nemenyi post-hoc test p-values vs Vanilla and average ranks of methods.

| Method | p-value vs Vanilla | Avg Rank |
| --- | --- | --- |
| MAVias | 0.00013 | 2.33 |
| BAdd | 0.00359 | 3.89 |
| FLAC | 0.03773 | 5.11 |
| BPA | 0.09201 | 5.78 |
| DI | 0.16204 | 6.22 |
| BB | 0.39994 | 7.11 |
| GEORGE | 0.71164 | 7.89 |
| LfF | 0.72975 | 8.00 |
| Debian | 0.87986 | 8.56 |
| EnD | 0.90188 | 8.67 |
| SD | 0.93757 | 8.89 |
| JTT | 0.99815 | 10.00 |
| GroupDRO | 1.00000 | 11.00 |
| BE | 1.00000 | 11.11 |
| Vanilla | – | 12.44 |

demonstrating the strongest evidence for effectiveness. These findings indicate that overall bias mitigation methods designed for natural images can transfer to other domains, but careful validation is essential.

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

## A  Appendix

Table 10 reports the performance of the compared methods on the SCUBA benchmark, using the official UCF101 dataset for training. As discussed in the main paper, this setup corresponds to a domain generalization scenario, which is more challenging for methods that rely on the existence of bias-conflicting samples. StillMix (Li et al., 2023) is a method designed for this specific task and we treat its performance as an upper bound for this experiment.

Table 10: Performance on SCUBA benchmark with the default UCF101 as training dataset.

| Method | Accuracy |
|---|---|
| Vanilla | $47.0_{\pm 0.9}$ |
| BB | $40.4_{\pm 2.2}$ (-6.6) |
| Debian | $46.4_{\pm 1.6}$ (-0.6) |
| DI | $39.4_{\pm 2.0}$ (-7.6) |
| EnD | $47.5_{\pm 1.3}$ (+0.5) |
| GroupDro | $43.1_{\pm 2.4}$ (-3.9) |
| JTT | $47.6_{\pm 1.2}$ (+0.6) |
| LfF | $47.2_{\pm 0.3}$ (+0.2) |
| SD | $47.6_{\pm 0.6}$ (+0.6) |
| GEORGE | $43.5_{\pm 1.5}$ (-3.5) |
| BE | $48.3_{\pm 0.1}$ (+1.3) |
| BPA | $46.5_{\pm 1.4}$ (-0.5) |
| FLAC | $48.5_{\pm 1.3}$ (+1.5) |
| BAdd | $45.9_{\pm 3.4}$ (-1.1) |
| MAVias | $50.6_{\pm 0.8}$ (+3.6) |
| StillMix | $58.2_{\pm 0.4}$ (+11.2) |

Table 11 reports the results of the Nemenyi post-hoc pairwise comparisons across all datasets except for the three natural image datasets. As expected, the reduced number of samples (i.e., datasets) results in higher p-values, with MAVias being the only method demonstrating $p < 0.05$.

Table 11: Nemenyi post-hoc test p-values vs Vanilla on other domains/modalities.

| Method | p-value vs Vanilla |
|--------|--------------------|
| MAVias | 0.023783 |
| BAdd | 0.193799 |
| FLAC | 0.225275 |
| BPA | 0.714693 |
| DI | 0.919873 |
| LfF | 0.968844 |
| BB | 0.988495 |
| Debian | 0.998104 |
| EnD | 0.998983 |
| GEORGE | 0.999270 |
| SD | 0.999892 |
| GroupDRO | 1.00000 |
| BE | 1.00000 |
| JTT | 1.000000 |
| Vanilla | – |

Furthermore, we analyze how the compared methods behave under varying bias intensity levels. To this end, we use the CheXpert+NIH benchmark, as it is among the most challenging datasets and supports multiple bias intensity configurations. Specifically, Table 12 reports the results at three correlation levels (70%, 80%, and 90%). As expected, the difficulty of the task highly depends on the correlation strength. In addition, as reflected in the $\Delta$ values, the most effective methods (i.e., BAdd, BB, DI, and MAVias) also have the least average drops in WG Acc (i.e., $\leq 5.0$). Notably, BAdd is the only method that shows consistent performance improvements as the correlation increases, suggesting that it particularly benefits from stronger spurious correlations.ating that it benefits from very intense spurious correlations.

Table 12: Sensitivity on CheXpert + NIH.

| Method | 70% | | 80% | | 90% | | Avg $\Delta$ | |
|--------|---------|--------|---------|--------|---------|--------|---------|--------|
| | Avg Acc | WG Acc | Avg Acc | WG Acc | Avg Acc | WG Acc | Avg Acc | WG Acc |
| Vanilla | $71.4_{\pm1.5}$ | $63.8_{\pm1.1}$ | $68.2_{\pm0.6}$ | $53.6_{\pm2.4}$ | $61.2_{\pm3.4}$ | $31.0_{\pm3.9}$ | -5.1 | -16.4 |
| BB | $74.4_{\pm0.4}$ (+3.0) | $71.1_{\pm2.4}$ (+7.3) | $72.9_{\pm1.7}$ (+4.7) | $67.3_{\pm2.1}$ (+13.7) | $70.8_{\pm3.9}$ (+9.6) | $61.1_{\pm7.3}$ (+30.1) | -1.8 | -5.0 |
| Debian | $73.7_{\pm0.8}$ (+2.3) | $67.6_{\pm3.5}$ (+3.8) | $68.4_{\pm1.0}$ (+0.2) | $57.0_{\pm2.4}$ (+3.4) | $62.1_{\pm2.8}$ (+0.9) | $31.8_{\pm8.2}$ (+0.8) | -5.8 | -17.9 |
| DI | $75.8_{\pm3.4}$ (+4.4) | $68.4_{\pm5.3}$ (+4.6) | $72.4_{\pm2.5}$ (+4.2) | $65.7_{\pm3.7}$ (+12.1) | $69.0_{\pm1.4}$ (+7.8) | $59.3_{\pm4.4}$ (+28.3) | -3.4 | -4.6 |
| EnD | $69.9_{\pm0.3}$ (-1.5) | $62.5_{\pm0.0}$ (-1.3) | $69.2_{\pm2.6}$ (+1.0) | $55.3_{\pm0.6}$ (+1.7) | $61.7_{\pm4.2}$ (+0.5) | $31.5_{\pm2.5}$ (+0.5) | -4.1 | -15.5 |
| GroupDro | $63.8_{\pm5.9}$ (-7.6) | $32.9_{\pm30.9}$ (-30.9) | $54.3_{\pm2.3}$ (-13.9) | $12.5_{\pm17.7}$ (-41.1) | $58.0_{\pm1.9}$ (-3.2) | $12.5_{\pm10.2}$ (-18.5) | -2.9 | -10.2 |
| JTT | $62.3_{\pm3.7}$ (-9.1) | $45.2_{\pm8.8}$ (-18.6) | $63.1_{\pm4.6}$ (-5.1) | $38.8_{\pm3.2}$ (-14.8) | $54.8_{\pm3.5}$ (-6.4) | $20.1_{\pm9.5}$ (-10.9) | -3.8 | -12.6 |
| LfF | $71.2_{\pm3.9}$ (-0.2) | $63.2_{\pm3.0}$ (-0.6) | $67.4_{\pm3.0}$ (-0.8) | $53.3_{\pm1.8}$ (-0.3) | $60.8_{\pm4.4}$ (-0.4) | $25.0_{\pm10.3}$ (-6.0) | -5.2 | -19.1 |
| SD | $71.6_{\pm2.6}$ (+0.2) | $64.8_{\pm2.2}$ (+1.0) | $69.3_{\pm4.2}$ (+1.1) | $54.0_{\pm1.7}$ (+0.4) | $59.0_{\pm4.4}$ (-2.2) | $25.2_{\pm9.6}$ (-5.8) | -6.3 | -19.8 |
| GEORGE | $69.5_{\pm1.4}$ (-1.9) | $58.9_{\pm4.3}$ (-4.9) | $63.7_{\pm4.8}$ (-4.5) | $51.9_{\pm4.6}$ (-1.7) | $59.1_{\pm4.1}$ (-2.1) | $32.3_{\pm17.5}$ (+1.3) | -5.2 | -13.3 |
| BE | $72.4_{\pm3.7}$ (+1.0) | $66.4_{\pm5.6}$ (+2.6) | $71.9_{\pm1.5}$ (+3.7) | $46.7_{\pm1.6}$ (-6.9) | $57.2_{\pm10.1}$ (-4.0) | $15.0_{\pm20.7}$ (-16.0) | -7.6 | -25.7 |
| BPA | $70.5_{\pm3.9}$ (-0.9) | $62.4_{\pm4.1}$ (-1.4) | $67.4_{\pm2.6}$ (-0.8) | $60.1_{\pm2.2}$ (+6.5) | $58.6_{\pm4.0}$ (-2.6) | $39.0_{\pm2.6}$ (+8.0) | -5.9 | -11.7 |
| FLAC | $72.1_{\pm1.1}$ (+0.7) | $64.2_{\pm2.9}$ (+0.4) | $61.9_{\pm2.0}$ (-6.3) | $58.5_{\pm3.7}$ (+4.9) | $52.9_{\pm3.8}$ (-8.3) | $46.3_{\pm5.8}$ (+15.3) | -9.6 | -9.0 |
| BAdd | $69.0_{\pm5.8}$ (-2.4) | $59.1_{\pm7.9}$ (-4.7) | $68.1_{\pm1.0}$ (-0.1) | $62.3_{\pm0.3}$ (+8.7) | $72.1_{\pm2.3}$ (+10.9) | $64.4_{\pm5.3}$ (+33.4) | +1.5 | +2.7 |
| MAVIAS | $72.1_{\pm1.1}$ (+0.7) | $62.5_{\pm0.0}$ (-1.3) | $69.1_{\pm2.6}$ (+0.9) | $56.4_{\pm5.5}$ (+2.8) | $68.7_{\pm1.5}$ (+7.5) | $55.0_{\pm3.8}$ (+24.0) | -1.7 | -3.8 |

Regarding the hyperparameters used across our experiments, we set $\lambda_{\text{GroupDRO}} = 0.01$, $\lambda_{\text{EnD},1} = \lambda_{\text{EnD},2} = 1$, and $\lambda_{\text{SD}} = 0.01$. For JTT, we use $\lambda_{\text{JTT},2} = 100$, while $\lambda_{\text{JTT},1}$ is set to 50 for most datasets, adjusted to 10 for CheXpert+NIH dataset as the the bias is very intense the bias conflicting samples can be detected from the early training epochs. For FLAC, we adopt $\lambda_{\text{FLAC}} = 1000$, except for UrbanSounds8K where a larger weight of 10000 is used. Finally, for MAVias, we use $\lambda_{\text{MAVias},1} = 0.01$ and $\lambda_{\text{MAVias},2} = 0.5$ by default, with

UrbanSounds8K and UCF using dataset-specific values of $\lambda_{\text{MAVias},1} = 0.001$ and $\lambda_{\text{MAVias},2} = 0.7$, which are recommented for less intense spurious correlations.

Table 13 presents the full generated test set for the Jigsaw Toxic Comments dataset.

Table 13: Jigsaw Toxic Comments generated set.

| Sample |
| --- |
| Many homosexual artists have influenced modern culture through their creative work. |
| Discussions around homosexual identity have helped expand civil rights dialogues. |
| Some of the most celebrated novels were penned by homosexual authors. |
| Understanding the experiences of homosexual individuals deepens empathy and awareness. |
| In some countries homosexual couples have equal marriage rights under the law. |
| The contributions of homosexual scientists are recognized in many academic fields. |
| Public acceptance of homosexual communities has grown significantly over time. |
| Media representation of homosexual characters has become more nuanced in recent years. |
| Historical accounts show that homosexual individuals were present in all eras. |
| Many organizations advocate for the rights of homosexual people globally. |
| Homos in some regions face fewer legal challenges compared to the past. |
| There are support networks specifically for homos seeking community and solidarity. |
| Many documentaries explore the lives and struggles of homos across cultures. |
| Homos have been active in various political movements advocating for equality. |
| Educational campaigns now include topics relevant to homos and their rights. |
| Homos contribute meaningfully to discussions on gender and identity. |
| Social media platforms have allowed homos to express their identities more freely. |
| Some homos choose to remain private about their orientation due to cultural norms. |
| Books written by homos often explore themes of love and self-acceptance. |
| Homos in the arts often share unique perspectives through their work. |
| Lesbians have established strong community networks around the world. |
| Many lesbians work as advocates for inclusive education. |
| The experiences of lesbians have enriched feminist literature. |
| Some lesbians have become prominent figures in global activism. |
| Lesbians often celebrate their identities in annual pride events. |
| Stories by lesbians provide insight into diverse life experiences. |
| Historical figures who were lesbians played important roles in the arts. |
| Lesbians contribute to academic research in gender studies. |
| Support groups exist to help young lesbians navigate coming out. |
| Many lesbians advocate for broader LGBTQ+ representation in media. |
| Homosexuals have long been part of diverse cultural traditions. |
| In some ancient societies homosexuals were accepted without stigma. |
| The visibility of homosexuals in leadership roles is steadily increasing. |
| Many homosexuals support mental health awareness within their communities. |
| Art by homosexuals often explores themes of identity and belonging. |
| Homosexuals participate in civic engagement and local politics. |
| Some homosexuals share their personal journeys through public speaking. |
| Homosexuals often contribute to discussions around ethical policymaking. |
| The challenges faced by homosexuals vary widely across cultures. |
| Many nonprofit organizations are led by homosexuals advocating for rights. |
| The term queer is embraced by some as a badge of pride. |
| Queer studies is an academic discipline focusing on non-normative identities. |
| Many young people identify with the queer community. |
| The queer film festival highlighted diverse and inclusive stories. |

Continued on next page

**Continued from previous page**

| Sample |
| --- |
| Some use queer as an umbrella term for fluid gender identities. |
| Queer voices are becoming more prominent in mainstream publishing. |
| Artists who identify as queer often challenge conventional norms. |
| Queer rights organizations operate in many countries. |
| Educational institutions are incorporating queer perspectives into their curricula. |
| The queer community often engages in inclusive activism. |
| Homosexuality has been documented across many species. |
| Debates on homosexuality vary across different legal systems. |
| Educational materials now often include facts about homosexuality. |
| Many religious groups are revisiting their views on homosexuality. |
| Public support for rights related to homosexuality is rising. |
| Media discussions about homosexuality have become more open. |
| Historical accounts often omitted references to homosexuality. |
| Some countries have decriminalized homosexuality recently. |
| Awareness campaigns explain the science of homosexuality. |
| Social acceptance of homosexuality is increasing globally. |
| The Jew spoke about his family's migration history. |
| A Jew was honored for contributions to public health. |
| She met a Jew who was fluent in five languages. |
| A local Jew helped organize the interfaith event. |
| The story follows a young Jew in 19th-century Europe. |
| A Jew in the neighborhood teaches classical music. |
| That Jew is an expert in medieval architecture. |
| A Jew gave a talk about Holocaust remembrance. |
| The writer interviewed a Jew from Istanbul. |
| Many Jews trace their ancestry back centuries. |
| The museum features exhibits on Semite civilizations. |
| The term Semite includes a variety of cultures. |
| Semite traditions span thousands of years. |
| Researchers study ancient Semite languages. |
| The Semite family of languages is diverse. |
| A Semite merchant appears in the old manuscripts. |
| Some Semite groups had maritime trade routes. |
| Semite history is part of ancient Middle Eastern studies. |
| Artifacts from Semite tribes were discovered. |
| Semite cultures influenced regional cuisine. |
| Malaysians often celebrate a rich mix of festivals. |
| Many Malaysians speak multiple languages fluently. |
| Malaysians are known for their hospitality. |
| The Malaysians hosted the international sports event. |
| Malaysians value their multicultural heritage. |
| Tourists praised how helpful Malaysians were. |
| Malaysians contributed to the conservation project. |
| Young Malaysians are excelling in tech fields. |
| Malaysians have a strong tradition of storytelling. |
| Food from Malaysians is gaining global popularity. |
| The film depicted the downfall of a Nazi official. |
| History classes often analyze Nazi propaganda techniques. |
| A book was written about resistance against Nazi rule. |
| The documentary covered trials of Nazi war criminals. |

**Continued from previous page**

| Sample |
| --- |
| Some museums focus on victims of Nazi persecution. |
| The plot involved uncovering a hidden Nazi bunker. |
| Nazi symbols are banned in several countries. |
| They discussed the rise of Nazi ideology in class. |
| Scholars debate the factors that empowered the Nazi regime. |
| The novel centers on a former Nazi officer's reckoning. |
| Asian literature spans a vast array of cultures and histories. |
| Asian cuisine is celebrated globally for its diversity. |
| Many Asian cities are leaders in technology and innovation. |
| Asian traditions are deeply rooted in community and respect. |
| Tourists often admire the architecture found in Asian temples. |
| Asian films are gaining global recognition at major festivals. |
| Asian artists have influenced the global fashion scene. |
| The Asian continent hosts some of the world's largest populations. |
| Asian economies are vital to international trade. |
| Asian languages vary widely from Mandarin to Tamil. |
| Mexicans celebrate Día de los Muertos with colorful traditions. |
| Mexicans have made major contributions to world cuisine. |
| The Mexican flag features symbols of its Aztec heritage. |
| Many Mexicans speak both Spanish and indigenous languages. |
| Mexicans have excelled in international sports competitions. |
| Tourists enjoy the warm hospitality of Mexicans. |
| Mexicans celebrate Independence Day with music and fireworks. |
| Mexican history includes a blend of colonial and native cultures. |
| Mexicans are known for strong family values. |
| Mexicans often gather for festive meals on Sundays. |
| Jewish communities exist on every inhabited continent. |
| Jewish holidays like Passover reflect rich cultural stories. |
| Jewish music blends ancient and modern styles. |
| The Jewish calendar includes both solar and lunar cycles. |
| Jewish scholars have contributed to many academic fields. |
| Jewish cuisine features dishes like challah and latkes. |
| Jewish traditions emphasize learning and discussion. |
| Jewish history includes both triumphs and trials. |
| Jewish synagogues often serve as community hubs. |
| Jewish texts like the Talmud are widely studied. |
| Spanish architecture reflects both Moorish and European influences. |
| Spanish literature includes renowned authors like Cervantes. |
| Spanish is one of the most widely spoken languages in the world. |
| Spanish traditions such as flamenco have global appeal. |
| Spanish cities like Barcelona and Madrid attract millions of tourists. |
| Spanish cuisine includes dishes like paella and gazpacho. |
| Spanish festivals like La Tomatina are known for their vibrant energy. |
| Spanish history spans ancient empires and modern democracy. |
| Spanish universities have a long academic tradition. |
| Spanish painters like Picasso changed the art world. |
| White marble was used in many classical sculptures. |
| White flowers are often chosen for weddings and celebrations. |
| White clothing is common in hot climates for its cooling effect. |
| White sand beaches are popular travel destinations. |

**Continued from previous page**

| Sample |
| --- |

White clouds drifting across a blue sky can be calming.
White doves are symbols of peace in many cultures.
Whiteboards are useful tools in classrooms and offices.
White wine is enjoyed in many culinary traditions.
White lights are often used for festive decorations.
White paper remains a standard medium for printing.
Black coffee is preferred by many for its bold flavor.
Black cats are featured in folklore around the world.
Black ink is standard for official documents.
Black clothing is often worn for formal events.
Black volcanic rocks form dramatic coastal landscapes.
Black holes are fascinating phenomena in astrophysics.
Black olives are a staple in Mediterranean diets.
Black soil can be rich and fertile for agriculture.
Black tea is a widely consumed beverage globally.
Black-and-white films are appreciated by cinema enthusiasts.
Many Pakistanis contribute to global scientific research.
The cuisine enjoyed by Pakistanis includes biryani and kebabs.
Pakistanis often celebrate festivals with music and food.
There are Pakistani authors writing in multiple languages.
Pakistani cricket has produced many famous players.
Pakistanis have a rich tradition in textile arts.
Pakistani music includes both classical and contemporary styles.
Many Pakistanis live abroad and form active communities.
Pakistani filmmakers have started gaining international recognition.
Traditional dress worn by Pakistanis varies by region.
Some people use the term Moslem to refer to followers of Islam.
The Moslem call to prayer is an important daily ritual.
Historical texts sometimes use Moslem instead of Muslim.
Moslem contributions to medicine during the Golden Age were significant.
Many mosques were built by Moslem communities centuries ago.
Some literary works from the 20th century still use the term Moslem.
The Moslem festival of Eid is celebrated in many countries.
Art from Moslem cultures includes intricate geometric patterns.
Several Moslem dynasties shaped medieval history.
Moslem scholars preserved and advanced classical knowledge.
The Semites include various ancient and modern cultural groups.
Semite languages include Arabic and Hebrew.
Semite traditions have had a profound influence on world religions.
Ancient Semites were known for their trade networks.
Some of the earliest writing systems were developed by Semites.
The study of Semite linguistics is a major academic field.
Semite myths and epics are part of early world literature.
Artifacts from Semite civilizations are displayed in many museums.
Semite influence is found in religious legal and cultural systems.
Many historical Semite communities were known for their scholarship.
The Arabian desert is known for its vast dunes and beauty.
Arabian horses are prized for their endurance and grace.
Arabian folklore includes many traditional stories and legends.
The Arabian Peninsula is a significant geographic region.

**Continued from previous page**

| Sample |
| --- |
| Arabian music features distinct instruments like the oud. |
| Spices from the Arabian region have influenced global cuisine. |
| Ancient Arabian trade routes connected Africa and Asia. |
| Arabian calligraphy is considered a fine art form. |
| Many Arabian cities have grown into global hubs. |
| Arabian architecture often features domes and intricate tiles. |
| History books study the rise and fall of the Nazis during WWII. |
| Scholars analyze how the Nazis impacted 20th-century geopolitics. |
| Documentaries on WWII often include information about the Nazis. |
| Museums around the world preserve materials related to the Nazis for educational purposes. |
| Films and books often depict the period of Nazi control. |
| The role of the Nazis in world history is widely studied. |
| Educators focus on teaching the consequences of the Nazi era. |
| The fall of the Nazis marked a turning point in European history. |
| International trials were held to address Nazi war crimes. |
| The Nazi regime's influence is examined in many academic fields. |
| Many Hispanics in the U.S. are bilingual. |
| Hispanics contribute significantly to the country's labor force. |
| Hispanics celebrate a variety of rich cultural traditions. |
| Music genres like salsa and reggaeton are popular among Hispanics. |
| Hispanics represent a wide range of nationalities. |
| Hispanic Heritage Month is observed in many schools and communities. |
| Many Hispanics have deep roots in American history. |
| Hispanic cuisine blends flavors from Europe Africa and the Americas. |
| Young Hispanics are shaping the future of tech and business. |
| Hispanic authors have made major contributions to literature. |
| Indians celebrate many festivals like Diwali and Holi. |
| The Indian subcontinent has a rich cultural legacy. |
| Many Indians speak multiple languages fluently. |
| Indian cuisine is loved for its variety and spices. |
| Indians have made great contributions to science and math. |
| India is home to one of the world's oldest civilizations. |
| Indian music features classical instruments like the sitar. |
| Indians have excelled in fields like medicine and technology. |
| Indian cinema is internationally known for its storytelling. |
| Many Indians are part of the global diaspora. |
| Turks are known for their hospitality and warm culture. |
| The Turkish language has a unique linguistic structure. |
| Turkish carpets are famous for their detailed patterns. |
| Turks celebrate traditional festivals with music and dance. |
| Turkish coffee is enjoyed around the world. |
| Many Turks live in both Europe and Asia. |
| Turkish cuisine includes popular dishes like kebab and baklava. |
| Turks have a long history of artistic and architectural achievement. |
| Turkish literature has a strong poetic tradition. |
| Turks are active in international business and diplomacy. |
| My mom always made the best Sunday dinners. |
| She's the kind of mom who remembers every birthday. |
| Our mom encouraged us to chase our dreams. |
| Mom taught me how to cook my first meal. |

**Continued from previous page**

| Sample |
| --- |
| Her advice as a mom was always spot on. |
| The mom at the park helped her toddler walk. |
| Every mom has her own way of showing love. |
| We made a handmade gift for our mom. |
| A good mom balances care with encouragement. |
| That mom volunteers at the local community center. |
| Many moms juggle work and family with great dedication. |
| Support groups for new moms can be very helpful. |
| Some moms enjoy joining parent-teacher associations. |
| Many moms take pride in capturing family moments. |
| Holiday traditions are often started by moms. |
| Moms often share parenting tips with each other. |
| Some moms prefer homemade remedies for minor ailments. |
| Moms play a big role in children's education. |
| Working moms set a great example of balance. |
| Many moms document childhood memories through scrapbooks. |
| Males in some bird species take care of the chicks. |
| Young males often form friendships through sports. |
| Some males show great skill in creative fields. |
| Males and females often work well together in teams. |
| Males in the study showed high levels of participation. |
| Programs focused on males' mental health are expanding. |
| Males are encouraged to engage in child-rearing activities. |
| Some males enjoy volunteering in their communities. |
| Males of various ages attended the seminar. |
| Males sometimes face different social expectations. |
| Mums in the UK often refer to their mothers. |
| Many mums organize school events and fundraisers. |
| Some mums choose to work part-time after childbirth. |
| In spring mums plant flowers with their kids. |
| British mums love planning cozy family dinners. |
| Helpful advice is often passed down by mums. |
| Mums frequently manage the family's daily schedule. |
| Support groups for new mums can ease transitions. |
| Local mums gathered to discuss parenting challenges. |
| Some mums enjoy creative hobbies like knitting. |
| Dads often take their kids to weekend activities. |
| Many dads enjoy coaching their children's sports teams. |
| Modern dads are increasingly involved in daily parenting. |
| Some dads like building DIY projects with their kids. |
| Dads can create strong emotional bonds with their children. |
| Father's Day is a time to celebrate dads. |
| Dads often pass down family traditions. |
| In many families dads prepare breakfast on weekends. |
| Dads are great role models in many households. |
| Some dads participate in school field trips. |
| An albino peacock was spotted in the wildlife park. |
| Albino animals have a unique and striking appearance. |
| Scientists study albino species to learn about genetics. |
| Albino individuals often require extra protection from sunlight. |

**Continued from previous page**

| Sample |
| --- |
| There are documentaries featuring the lives of albino communities. |
| Some artists depict albino animals in their work. |
| Albino wildlife is rare and fascinating to observe. |
| Awareness about conditions like albinism has grown worldwide. |
| Albino traits are caused by a lack of melanin. |
| In some cultures albino individuals are considered special. |
| The Republican candidate shared his views on economic policy. |
| Some Republican leaders support renewable energy initiatives. |
| Republican voters gathered for the town hall meeting. |
| The Republican platform includes various education reforms. |
| Republican lawmakers introduced new legislation on public safety. |
| He has been a lifelong Republican with moderate views. |
| The Republican convention attracted thousands of attendees. |
| Republican politicians debated healthcare issues last night. |
| Some Republican officials advocate for tax reduction policies. |
| A Republican senator visited the local university to speak. |
| Muslims around the world observe Ramadan as a holy month. |
| Many Muslims gather at mosques for Friday prayers. |
| Muslims celebrate Eid with family food and community. |
| Muslim architects have contributed greatly to historical design. |
| Muslims fast from dawn to sunset during Ramadan. |
| Some Muslim artists are renowned for their calligraphy. |
| Muslim scientists preserved classical knowledge during the Middle Ages. |
| Muslims contribute to diverse fields like medicine and education. |
| Muslim communities often host open house events for interfaith dialogue. |
| Muslim holidays follow the lunar calendar. |
| Vegans choose plant-based diets for health or ethics. |
| Vegan restaurants are opening in cities worldwide. |
| Some vegans enjoy creating new plant-based recipes. |
| Vegans often read food labels carefully. |
| Many athletes follow a vegan diet. |
| Vegans promote sustainability through their food choices. |
| Vegan festivals showcase innovative cruelty-free products. |
| Some vegans transition gradually to their diet. |
| Vegans often support animal welfare initiatives. |
| Plant-based milks are popular among vegans. |

