# OpenReview forum: "Do Bias Mitigation Methods Generalize? A Cross-Modality Study"
_TMLR — Rejected by TMLR_

### Review · Reviewer_DXem · 2026-03-11

**Summary Of Contributions:**

**Summary**

This work presents an evaluation of a selection of model debiasing methods conceived for natural images across different modalities, namely text, audio, medical images and videos. The authors focus on three research questions including: (i) the capability of such methods to mitigate bias across different modalities (and as such, improving generalization over ERM vanilla models), (ii) if the ranking in terms of mitigation performance established with respect to natural images is preserved across modalities, and finally whether these potential improvements are statistically significant.  Experiments with 11 methods over 6 datasets are reported and empirically suggest positive responses to the posed questions.


**Strengths**


The paper is overall well-written and easy to follow. The problem of mitigating bias is of general interest, with an increasingly large number of contributions in the last few years.



**Weaknesses**


-	I believe that the objective of the paper could be better specified and framed. Specifically, at some point it is mentioned that the objective is to evaluate visual bias mitigation approaches beyond natural images. However, methods that are actually tailored to mitigate visual bias (e.g., methods based on generative or vision-language models) are not directly transferrable to other modalities, by definition. As such, I would suggest reframing the aim towards the evaluation of debiasing general strategies already benchmarked for visual bias to other modalities. This connects to my next two comments.
-	A major concern relies on the criterion employed for selecting the debiasing methods to be evaluated across modalities, which I believe makes complicated to get general insights on the reasons behind the failure or success in generalization. This point is critical as impacts the main contributions of the paper. Currently, debiasing methods are selected according to two different criteria: (i) seminal methods including JTT, LfF, GroupDRO, EnD, BB and DebiAN, and (ii) recent methods that show state-of-the-art performance. First, I believe that the list is not comprehensive, especially for the recent works, as there are only three methods included, and the criterion for their selection is not clearly defined. Furthermore, I believe that the selection should be made considering specific characteristics of the debasing protocol and mechanism, so that more insights could be provided regarding why some methods may or may not generalize across modalities, possibly being of more interest for the community.
-	I have doubts on the novelty and the impact of the proposed analysis. In principle, evaluating methods proposed for bias mitigation on natural images and other modalities is not novel, and different works already test debiasing methods on text and images. For instance, in [1, 2] the proposed methods are tested on visual images and text with experiments on CivilComments and MultiNLI, and a similar analysis is also found in the original publication introducing JTT. In general, methods are shown to be consistent in terms of generality improvements across the two different modalities. This also applies to the generalization of bias mitigation methods to medical images (e.g., in [1] or in the original publication introducing EnD).
-	Regarding medical imaging, I believe that evaluating the generality of bias mitigation to the medical domain would require the inclusion of several imaging modalities, considering not only the bias related to the acquisition systems, but also data collections protocols (e.g., demographic attributes) and bias across imaging modalities. Even if I agree that testing the generality to bias in the medical world is very interesting, that would require including more datasets and evaluating more types of bias.
-	In the background section, the selected methods are described with many details, however, I believe that the paper could benefit of a related work section where the current state-of-the-art on model debiasing mitigation methods is introduced and described, reporting a categorization with respect to the debiasing protocols, with a finer level of details than just bias-supervised and unsupervised.
-	I appreciate the statistical analysis of the obtained results. However, I believe that the third research question of the paper (proving statistical robustness in bias mitigation) should be considered a fundamental analysis to prove the second and the first contributions rather than an independent one, as one might argue that if the increase in generalization with respect to ERM is not statistically significant, then the explored methods are not actually capable of generalizing across modalities.

**References**

[1] Tiwari, R., Sivasubramanian, D., Mekala, A., Ramakrishnan, G., & Shenoy, P. (2024). Using early readouts to mediate featural bias in distillation. In Proceedings of the IEEE/CVF Winter Conference on Applications of Computer Vision (pp. 2638-2647).

[2] Izmailov, P., Kirichenko, P., Gruver, N., & Wilson, A. G. (2022). On feature learning in the presence of spurious correlations. Advances in Neural Information Processing Systems, 35, 38516-38532.

**Audience:**

Yes

**Audience Explanation:**

The problem of bias mitigation is of broad interest, with a growing research community working on designing robust debiasing methods. The evaluation of generalization across modalities, if deepened and expanded, could be of interest to the TMLR audience.

**Claims And Evidence:**

No

**Claims Explanation:**

In the current version, the selection of debiasing methods is not comprehensive. First, it only includes three methods published after 2024. Second, the criterion used for such selection is not sufficiently convincing, as it does not consider the debiasing mechanism, which limits the insights provided by this work. Furthermore, the evaluation across modalities involves only six datasets, which appears limited, especially for medical imaging.

**Requested Changes:**

-	The major revision would require expanding the methods selected for evaluation. First, including more recent methods (possibly with different debiasing mechanisms), then focusing on different debiasing protocols, to give more insights on the reasons behind failure or success to generalization to other modalities. This is particularly critical, as the generalization to other modalities (at least text, or medical imaging) is a protocol already employed in other works, with methods generally shown to maintain their debiasing capabilities across modalities. As such, I think that going deeper in the analysis, considering the specific debiasing mechanism would be necessary to be more of interest for the community. To provide an example, indicating a not comprehensive and purely indicative list of potential methods to include, it could be possible to consider two-step methods based on bias identification as a first step, possibly including approaches that perform bis pseudo-labeling based on model misclassifications (as JTT, already included), loss values (e.g., Bias Ensemble [3]), gradients (e.g., GERNE [4]), analysis of the embeddings (e.g., George [5]), or methods based on group robust optimization (e.g., BPA [6]).
-	Furthermore, I would suggest reframing the objective of the work and the contributions (see my comments in the summary of contributions).
-	Finally,  the paper could benefit from a related work section, providing an actual description of current state-of-the-art on model debiasing, specifically mentioning works that already evaluate bias across at least text, natural and medical images, while describing the main debiasing strategies.
-	(Minor) Considering the contributions, I believe that the word “domains” could be replaced with “modalities” in the title.

**References**

[3] Lee, J., Park, J., Kim, D., Lee, J., Choi, E., & Choo, J. (2023, June). Revisiting the importance of amplifying bias for debiasing. In Proceedings of the AAAI conference on artificial intelligence (Vol. 37, No. 12, pp. 14974-14981).

[4] Asaad, I., Shadaydeh, M., & Denzler, J. (2025). Gradient extrapolation for debiased representation learning. In Proceedings of the IEEE/CVF International Conference on Computer Vision (pp. 3819-3829).

[5] Sohoni, N., Dunnmon, J., Angus, G., Gu, A., Ré, C.: No subclass left behind: Fine-grained robustness in coarse-grained classification problems. Advances in Neural Information Processing Systems 33, 19339–19352 (2020) 2, 4, 10

[6] Seo, S., Lee, J. Y., & Han, B. (2022). Unsupervised learning of debiased representations with pseudo-attributes. In Proceedings of the IEEE/CVF Conference on Computer Vision and Pattern Recognition (pp. 16742-16751).

---

> ### Author Response · Authors · 2026-04-14
> **Response to Reviewer DXem**
>
> We thank the reviewer for the thorough and constructive feedback. We appreciate the recognition of the clarity of the paper and the importance of the problem. Below, we respond to the reviewer’s concerns and summarize the revisions made in the revised manuscript.
>
> ### Methodologies and Categorization
>
> We appreciate this important point and have made several improvements in this direction.
>
> First, we expanded the set of considered methodologies by incorporating the methods suggested by the reviewer (Bias Ensemble, BPA, and GEORGE). Regarding GERNE, the code was not publicly available at the time we prepared the revision. In particular, the code was published a few days ago, and the revision timeline did not allow us to run the full set of experiments - we will include it in the final version of the paper. Also, if additional recent approaches have value to be involved in this analysis, we would be happy to integrate them as well.
>
> In addition, we reorganized the methods along two axes:
> (a) how the bias signal is derived, and
> (b) how the bias is mitigated.
>
> We updated the experimental analysis (Tables 7 and 8) to reflect this categorization and to better understand how different categories of methods behave across modalities.
>
> ### Objective of the Work and Contributions
>
> We agree that the original phrasing (“visual bias mitigation methods”) could be misleading. In the revised manuscript, we instead refer to “bias mitigation methods that are primarily evaluated on natural images”, which more accurately reflects our scope.
>
> We also added a new subsection, *“Cross-modality Evaluations”*, in the related work section. There, we explicitly discuss prior works that evaluate debiasing methods beyond vision (e.g., text and medical imaging), and we clarify how our work differs from and extends these efforts.
>
> Regarding medical imaging, we expanded Section 2.1.1 to include a discussion of demographic biases and the broader challenges specific to this domain. We clarify that our study focuses on a particular type of bias in medical images and provide justification for not including demographic bias in our analysis.
>
> Finally, we agree with the reviewer regarding RQ3. We have removed it as a standalone research question.
>
> ### Related Work
>
> Following the reviewer’s suggestions, we significantly expanded the related work section. We now provide a more structured overview of the bias mitigation literature, including a categorization based on methods’ characteristics. We also discuss in more detail which methods have already been evaluated across multiple modalities.
>
> ### Title
>
> We updated the title to:
> **“Do Bias Mitigation Methods Generalize? A Cross-Modality Study”**,
> removing “visual” and “domain” to better reflect the scope and contributions of the work.

---

### Review · Reviewer_tCTF · 2026-03-24

**Summary Of Contributions:**

This paper presents a comprehensive benchmark evaluating the cross-domain generalization of 11 visual bias mitigation methods across four distinct modalities: text, audio, medical imaging, and video. By testing these approaches on six diverse datasets, the authors provide empirical evidence that spurious correlations can be addressed using modality-agnostic techniques, filling a significant gap in current bias research. This study establishes a valuable benchmark resource for developing bias mitigation methods that extend beyond the natural images domain.

Through rigorous statistical testing using Friedman and Nemenyi post-hoc tests, the study identifies MAVias and BAdd as the most robust and consistent performers across all domains. Crucially, the paper also documents specific failure modes, such as the poor performance of the DI method in high-class video tasks and the struggles of GroupDRO with medical imaging artifacts. The technical correctness and the sheer scale of the experiments make this a highly valuable and reliable contribution to the machine learning community.

**Audience:**

Yes

**Audience Explanation:**

TMLR's audience will find these findings highly relevant as machine learning models are increasingly used in high-stakes, multi-modal applications. This paper provides the first systematic evidence that visual bias mitigation approaches can effectively transfer to text, audio, and medical domains, which is of significant interest to researchers seeking robust and trustworthy AI.

Practitioners in fields such as healthcare and NLP will specifically benefit from the identification of top-performing methods and the warnings regarding failure modes in complex data environments. By establishing a benchmark that extends beyond natural images, this work serves as an essential resource for developing more reliable, modality-agnostic solutions.

**Claims And Evidence:**

Yes

**Claims Explanation:**

The claims are supported by a massive benchmark involving 11 methods across 6 datasets and 4 modalities. This ensures the results reflect general cross-domain capabilities rather than specific dataset tuning. Formal statistical testing, including the Friedman and Nemenyi post-hoc tests, provides clear evidence for the effectiveness and ranking of methods like MAVias and BAdd.

The evidence is further strengthened by the transparent reporting of failure modes. By analyzing why certain methods struggled in medical imaging and high-class video tasks, the authors provide a realistic assessment of the current state of bias mitigation. This technical rigor makes the findings highly convincing for the machine learning community.

**Requested Changes:**

The submission is technically sound and the experimental evidence is sufficient to support the authors' claims as presented.

Minor comments:
- Analysis of Negative Transfer: Providing a brief, high-level discussion on why methods like GroupDRO or JTT might lead to performance degradation in the presence of complex medical imaging artifacts would offer deeper insights.
- MAVias Adaptation: A short reflection on whether MAVias's success in non-image domains is primarily driven by its mitigation loss ($L_\text{align}$) rather than its original image-tagging component would clarify its robustness for future researchers.

---

> ### Author Response · Authors · 2026-04-14
> **Response to Reviewer tCTF**
>
> We thank the reviewer for the positive and encouraging feedback. We are glad that the reviewer finds the benchmark comprehensive and the findings relevant to the community. We also appreciate the constructive suggestions for further improving the paper. Below, we respond to the minor comments.
>
> ### Analysis of Negative Transfer
>
> We agree that providing additional insight into failure cases would strengthen the paper. In the revised manuscript, we expand the Results section with method-specific discussions, where possible, to better explain observed failures.
>
> For example, JTT relies on the strong assumption that misclassified samples from a vanilla model correspond to bias-conflicting instances. In practice, this assumption does not consistently hold, leading to the inclusion of noisy or incorrectly identified samples, depending on the benchmark. Additionally, JTT applies uniform upweighting to all such samples, without accounting for varying levels of difficulty or noise. These factors contribute to its unstable behavior across different settings.
>
>
>
> ### MAVias Adaptation
>
> In the revised version, we include a more detailed discussion on the MAVias description, clarifying what components of MAVias are adapted.
>
> It is worth noting that the original image-tagging component is not used at all, thus MAVias’ success in non-natural image domains is indeed driven by its bias mitigation component.

---

### Review · Reviewer_X4VH · 2026-03-31

**Summary Of Contributions:**

## Summary:
The paper is a benchmark study of whether bias-mitigation methods developed for natural-image tasks also work in other modalities (cross-modality). Concretely, they evaluate 11 methods on six datasets, considering different domains such as text, audio, medical imaging, and video, with task-specific biased setups for each dataset. The paper also contributes a comparative ranking analysis across domains and a statistical significance analysis over datasets, rather than reporting isolated per-dataset results. Then, empirically, they show that transfer beyond vision is possible, but uneven: methods such as MAVias and BAdd are the most consistently strong, while others are much more domain-dependent. Overall, the work is positioned as a benchmark/resource paper meant to encourage the development of mitigation methods that generalize beyond standard vision benchmarks.

## strengths
- The paper addresses a clear and important gap: most prior bias-mitigation evaluations are vision-centric, whereas this study asks whether those methods generalize across modalities.
- They consider a range of evaluations, covering several modalities and multiple kinds of spurious correlations rather than a single benchmark.
- They go beyond raw scores by including method rankings and statistical testing, which makes the conclusions more convincing than a purely descriptive benchmark.
- The results are practically impactful: they identify methods that appear relatively robust across domains, especially MAVias and BAdd, and show that some popular methods are unstable outside their usual evaluation setting.

## weaknesses
- The major limitation is that this is still *a benchmark study* rather than a *mechanistic explanation of why some methods transfer and others do not*.
- Some evaluation choices are also somewhat constructed: For example: some bias settings are artificially induced, and for Jigsaw the paper builds a synthetic hard-negative test set with GPT-4, which is useful but may introduce its own artifacts.
- The modalities are diverse, but the benchmark remains small in the number of datasets, which the authors themselves note makes conservative statistical testing hard.
- There is also an adaptation issue for at least one method: MAVias’ original bias-detection stage is not applicable outside natural images, so the paper evaluates only its mitigation component with substituted bias-capturing features; that makes cross-method comparison slightly less clean.
- Since the study underlines broad comparison, some domain-specific implementation choices may affect the fairness of comparison across modalities.

**Audience:**

Yes

**Audience Explanation:**

The paper addresses a question likely relevant to a meaningful subset of the TMLR community: whether bias-mitigation methods developed for vision actually generalize across modalities. That is of interest to researchers working on robustness, fairness, spurious correlations, domain generalization, multimodal learning, and empirical benchmarking. Its value is not mainly in proposing a new algorithm, but in providing a systematic cross-domain comparison that can inform method selection and future research directions.

**Claims And Evidence:**

Yes

**Claims Explanation:**

- Yes, well mostly, but only for the paper’s benchmark-style empirical claims, not for any stronger causal or fully general claim.

- The evidence is clearest for the claim that visual bias-mitigation methods can transfer beyond natural images to some extent. The paper evaluates 11 methods across 6 datasets spanning text, audio, medical imaging, and video, with explicit task/spurious-attribute constructions and consistent fairness-oriented metrics such as worst-group accuracy or bias-conflicting-set accuracy. That is a reasonable experimental basis for the main empirical question.
- It is also fairly convincing that performance is method-dependent rather than universal. The tables show repeated gains for methods such as MAVias and BAdd, as well as clear failures or regressions for others, depending on the dataset. For example, on text and audio several methods improve substantially, while in medical imaging and video some methods degrade sharply; DI drops by 25.4 points on UCF101+SCUBA, and GroupDro drops by 18.5 points worst-group accuracy on CheXpert+NIH. That supports the paper’s more cautious conclusion that transfer is possible but must be validated per domain.
- All together; their claims are supported to a moderate-to-strong degree for an empirical benchmark paper: the evidence is clear, mostly convincing, and appropriately nuanced. But I would qualify that support with two reservations: some evaluations are synthetic/constructed, and the strongest method is partly adapted rather than tested in its original form. Therefore, the evidence supports qualified generalization claims, not broad universal ones!!

**Requested Changes:**

## Major
- I would ask authors pleaae discuss the fairness of comparing MAVias to the other methods. The paper notes that MAVias’ original bias-discovery component is not applicable outside natural images and is replaced with a different bias-capturing mechanism. This is important and should be discussed more prominently, including what exactly is preserved from the original method and what is changed, because it affects how readers interpret the strong MAVias results.
- Can you please strengthen the discussion of benchmark design limitations? Some datasets use injected spurious correlations, and Jigsaw relies on a GPT-4-generated hard-negative test set. These choices are reasonable for controlled evaluation, but the paper should more clearly discuss possible artifacts, external-validity limits, and how these design choices may influence rankings.
- and Plase Provide more details on statistical methodology and practical significance. The statistical testing is a strength, but the paper should more clearly explain the unit of comparison, the role of the added natural-image datasets in the significance analysis, and how readers should interpret non-significant pairwise differences given the small number of datasets.

## Minor:
- Please, if it is possible, include a more systematic sensitivity analysis. It would help to test whether the conclusions are stable to different correlation strengths, dataset subsampling choices, and hyperparameter settings, especially for the constructed-bias scenarios.
- Please separate “bias-label-aware” versus “bias-label-unaware” conclusions more carefully. The comparison is interesting, but it would be useful to discuss more explicitly when access to bias labels is a realistic assumption and how much of the performance gap may come from that supervision advantage.
- The paper usefully shows that some methods fail badly in certain domains. A short qualitative or quantitative analysis of why methods such as **DI, GroupDRO, or JTT** collapse in specific settings would make the benchmark more informative.

---

> ### Author Response · Authors · 2026-04-14
> **Respond to the Reviewer X4VH**
>
> We thank the reviewer for the thorough review. We appreciate the recognition of the paper’s strengths as a systematic benchmark, as well as the constructive feedback on how to further improve the manuscript. Below, we respond to the reviewer’s concerns and summarize the revisions made in the manuscript.
>
> ### Fairness of Comparing MAVias
>
> In the revised manuscript (Section 3.2 - MAVias description), we added a discussion to further clarify how MAVias is adapted and why this allows for fair comparison to the other relevant methods.
>
>
>
> ### Benchmark Design
>
> We agree that the benchmark design choices should be more explicitly discussed. In the revised manuscript:
>
> We involve the following discussion in Section 4.1 regarding the injected or constructed spurious correlations: “A subset of the considered datasets employs deliberately injected spurious correlations, where the training distribution is subsampled so that the spurious attribute co-occurs with the target label at a fixed rate. This form of controlled bias injection is a long-standing practice in the vision debiasing literature (Sagawa et al., 2019; Hong & Yang, 2021; Sarridis et al., 2024; 2025b) and characterizes most of the benchmarks on which the compared methods were originally validated (e.g., Waterbirds and UrbanCars). Its main benefit is that it isolates the effect of a single, well-defined spurious correlation at a known level. Therefore, adopting the same protocol in other modalities keeps our evaluation aligned with their original evaluation settings on images. Note that benchmarking on complex, unknown bias scenarios is beyond the scope of this work and constitutes an important direction for future research.”
>
> For the generated text set we clarify that this synthetic data is used only at the inference stage, thus any potential artifacts introduced by GPT-4 cannot influence the mitigation process of the compared methods.
>
>
> ### Statistical Analysis
>
> Clarifications on the statistical analysis to improve interpretability:
>
> - **Unit of comparison:** We treat each dataset as a single sample. This is now explicitly stated in the revised manuscript.
>
> - **Role of natural-image datasets:** The statistical analysis aims to assess the overall robustness of each method across all settings. Including natural-image datasets is therefore appropriate, as it both represents one of the application domains of the methods and increases the number of samples for the statistical tests. For completeness, in the revised version, we additionally provide a separate analysis in the Appendix restricted to non-natural image datasets only.
>
> - **Interpretation of results:** With a limited number of datasets, achieving statistical significance is inherently more challenging. As stated in the manuscript: *“Consequently, observing p < 0.05 is a strong indicator of robustness, while the absence of significance should not be interpreted as evidence that a method fails to reduce bias.”*
>
> ---
>
> ### Minor Comments
>
> **Sensitivity Analysis**
> While a full exploration is beyond the scope of the paper, we conducted a full set of experiments on the medical images for different correlation ratios; the results and a discussion on the behavior of the methods are presented in the Appendix (Table 12).
>
> **Bias-Label-Aware vs. Unaware Methods**
> In the revised version, the methods are classified in much more representative categories (both in terms of bias-label awareness and the mitigation mechanisms), and the results are updated accordingly.
>
> **Failure Mode Analysis**
> In the revised manuscript, we expand the Results section with method-specific discussions, where possible, to better explain observed failures.
>
> For example, JTT relies on the strong assumption that misclassified samples from a vanilla model correspond to bias-conflicting instances. In practice, this assumption does not consistently hold, leading to the inclusion of noisy or incorrectly identified samples, depending on the benchmark. Additionally, JTT applies uniform upweighting to all such samples, without accounting for varying levels of difficulty or noise. These factors contribute to its unstable behavior across different settings. Similarly, DI is sensitive to scenarios with a large number of bias-attribute classes and/or target classes, as it introduces additional classification heads and/or increases their complexity.

---

### Decision · Action_Editor_cyTM · 2026-05-31

**Recommendation:** Reject

**Audience:**

Yes

**Audience Explanation:**

Testing debiasing approaches in different modalities is an interesting question for the community.

**Claims And Evidence:**

No

**Claims Explanation:**

In this paper, the authors investigate whether bias mitigation methods, typically developed and benchmarked on natural images (such as CelebA or Waterbirds), actually generalize to other modalities. The motivation behind this analysis resides in the fact that spurious correlations, being predictive but non-causal shortcuts that models latch onto, are normally modality-agnostic and the methods designed to combat them have rarely been tested outside vision. The authors argue prior cross-modal work is fragmented and never spans more than two modalities at once, so they assemble what they call the first unified cross-modality benchmark. The proposed benchmark includes 14 methods evaluated on six datasets covering text, audio, medical imaging, and video.

The main findings is that most methods seem to transfer and reduce bias in at least some modalities, but consistency varies sharply, and per-domain validation should be addressed.

Overall the paper has improved his positioning and claims after the discussion phase - yet, while the mere empirical nature of the work is not per se a reason to not accept this work, the paper overclaims statistical relevance, including very large deviations in the results. Generality claims should also be reduced to the specific choice of dataset (unless strong motivation behind their choice is provided). Finally, the cohort of tested methods should include larger variety (groups of methods descend from the same family of debiasing approaches).

**Resubmission Of Major Revision:**

The authors may consider submitting a major revision at a later time.